# Extended Hierarchical Kriging Method for Aerodynamic Model Generation Incorporating Multiple Low-Fidelity Datasets

**Vinh Pham [1], Maxim Tyan [1] , Tuan Anh Nguyen [1] and Jae-Woo Lee [1,2,]***

1 Konkuk Aerospace Design-Airworthiness Institute (KADA), Konkuk University,
Seoul 05029, Republic of Korea; quangvinh@konkuk.ac.kr (V.P.); anhnt2407@konkuk.ac.kr (T.A.N.)
2 Department of Mechanical and Aerospace Engineering, Konkuk University, Seoul 05029, Republic of Korea
* Correspondence: jwlee@konkuk.ac.kr

**Abstract:** Multi-fidelity surrogate modeling (MFSM) methods are gaining recognition for their effectiveness in addressing simulation-based design challenges. Prior approaches have typically relied on recursive techniques, combining a limited number of high-fidelity (HF) samples with multiple low-fidelity (LF) datasets structured in hierarchical levels to generate a precise HF approximation model. However, challenges arise when dealing with non-level LF datasets, where the fidelity levels of LF models are indistinguishable across the design space. In such cases, conventional methods employing recursive frameworks may lead to inefficient LF dataset utilization and substantial computational costs. To address these challenges, this work proposes the extended hierarchical Kriging (EHK) method, designed to simultaneously incorporate multiple non-level LF datasets for improved HF model construction, regardless of minor differences in fidelity levels. This method leverages a unique Bayesian-based MFSM framework, simultaneously combining non-level LF models using scaling factors to construct a global trend model. During model processing, unknown scaling factors are implicitly estimated through hyperparameter optimization, resulting in minimal computational costs during model processing, regardless of the number of LF datasets integrated, while maintaining the necessary accuracy in the resulting HF model. The advantages of the proposed EHK method are validated against state-of-the-art MFSM methods through various analytical examples and an engineering case study involving the construction of an aerodynamic database for the KP-2 eVTOL aircraft under various flying conditions. The results demonstrated the superiority of the proposed method in terms of computational cost and accuracy when generating aerodynamic models from the given multi-fidelity datasets.

**Keywords:** multi-level multi-fidelity surrogate modeling; non-hierarchical low-fidelity data; extended co-Kriging; hierarchical Kriging; LRMFS



## 1. Introduction

Surrogate models have played an increasingly important role in different areas of aerospace engineering, such as aerodynamic data construction, aerodynamic shape optimization, structural design, multi-disciplinary optimization, and aircraft design, which require extensive physical tests or simulations. A precise and complete aerodynamic database is needed explicitly for aircraft design optimization and flight simulation to provide the aerodynamic coefficients of vehicles for various flying conditions and vehicle configurations throughout the entire mission. However, due to expensive and time-consuming computing, it is challenging to generate an enormous database using HF analysis methods like flight testing, wind tunnel testing, or Ansys Fluent RANS CFD simulation. Moreover, employing a simplified or LF model of the actual system may lead to reduced accuracy in the generated data. Therefore, surrogate models have emerged as effective solutions for constructing approximation models with limited HF data samples, enabling precise predictions of specific data points while maintaining cost efficiency.

Renowned single-fidelity surrogate modeling methods include polynomial response surface modeling (RSM) [1–3], radial basis function (RBF) neural networks [4,5], Kriging [6–8], and artificial neural networks (ANNs) [9–11]. They are often used to construct an approximation model from a single sampling set. The efficacy of surrogate models is notably contingent upon the density of training sample points. Converged accuracy in surrogate models demands a specific quantity of well-distributed training samples. However, acquiring an adequate number of training samples for precise surrogate fitting typically entails substantial costs, primarily driven by the necessity for conducting numerous tests or simulations. To address the challenge, prior studies [12–16] have focused on the development of an adaptive sampling strategy. This strategy aims to enhance the efficient construction of surrogate models in the context of surrogate-based design optimization problems, emphasizing the optimal selection of new data points over multiple iterations of model updates. Additionally, multi-fidelity surrogate models (MFSM), "variable fidelity models" (VFM), or "data fusion" are the next-level solutions for the problem, which allow for combining a small number of HF samples with lower-accuracy models or massive LF samples to generate an accurate approximation of an HF model. Here, the LF data can be exploited in significant quantities from simplified models or low-cost analyses.

In the realm of design optimization problems, numerous MFSM frameworks have emerged over the past two decades. These frameworks can be categorized into three popular types: bridge-function-based (BF), space mapping (SM), and Bayesian approaches. These methodologies have gained widespread acceptance within the engineering community [17,18]. First, in bridge-function-based approaches, the discrepancies between the HF and LF models are calibrated using bridge functions. Bridge functions can take the form of multiplicative, additive, or hybrid functions. For instance, Nguyen et al. [19] suggested using the trust region method with the modified variable complexity model (MVCM) to create a practical framework for interdisciplinary aircraft conceptual design. Both multiplicative and additive functions are constructed using neural network models in a comprehensive bridge-based framework. Tyan et al. [20] proposed global variable fidelity modeling (GVFM), in which the bridge functions are constructed using RBF models. The second approach, known as space mapping (SM), seeks the optimal conversion functions to map the design domain of the HF model to that of the LF model. Bandler et al. [21] first proposed the original SM method, which assumes a linear mapping between the inputs of the HF and LF models. Lastly, Kennedy and O'Hagan [22] and Quian et al. [23] developed the Bayesian-based MFSM frameworks, or co-Kriging models, allowing for additional flexibility, albeit the most sophisticated. Gratiet et al. [24] proposed a recursive co-Kriging model in which a fast cross-validation procedure was presented. Han et al. [25] proposed a hierarchical Kriging (HK) model, providing a faster and more efficient method to construct an MFSM than traditional co-Kriging. Jiang et al. [26] proposed a combination between the space mapping and Bayesian-based MFSM approaches, namely the space-mapping-based variable fidelity model (SM-VFM), in which a Gaussian process (GP) model is constructed for the LF model. Then, the VFM model is constructed by taking the predicted information from the LF model as prior knowledge and directly mapping it in the output space of the HF model. Tian et al. [27] proposed a transfer-learning-based variable-fidelity surrogate model (TL-VFM) for shell bulking prediction, which employs a two-stage training process to train deep neural networks (DNN) with multi-fidelity data. Xu et al. [28] employed the TL-VFM to develop a digital twin for a hierarchical stiffened plate. Meng et al. [13] proposed multi-fidelity deep neural networks (mDNNs) learning from multi-fidelity data to solve function approximation and inverse partial differential equation problems. Liu et al. [29] proposed a generative adversarial network for multi-fidelity data fusion (GAN-MDF) to develop a digital twin for a structured steel plate.

In order to reduce the number of HF simulations as much as possible, Le Gratiet [30] proposed multi-level co-Kriging (MCK) to recursively incorporate multiple datasets into the framework featuring a hierarchical scheme of fidelity levels. Han et al. [25,31] proposed a multi-level hierarchical Kriging (MHK) framework for efficient aerodynamic shape opti-

mization incorporating three or more levels of fidelity. Bu et al. [32] utilized MHK for an efficient aerostructure optimization of helicopter rotors toward aeroacoustic noise reduction. Pham et al. [33] proposed adaptive multi-fidelity data fusion strategy to incorporate multiple MFSM methods to efficiently handle data fusion problems featuring non-uniform aerodynamic data.

The majority of prior research within this list primarily involves MFSM methods that employ recursive MFSM frameworks tailored to multi-level datasets. These frameworks typically deal with fidelity levels that are absolutely distinguishable across the design space. For example, various aerodynamic study methodologies, such as wind tunnel experiments, 3D simulations, 2D simulations, panel methods, and empirical approaches, exhibit diverse fidelity levels that span the spectrum from high to low. However, there is another scenario to consider involving non-level LF datasets, where an exact delineation of fidelity levels across the entire design space is impractical. In these situations, it becomes highly advantageous to comprehensively harness the information from all LF datasets to construct the HF model. This scenario frequently arises in engineering design problems but has been relatively underexplored in the existing literature.

To address these challenges, we propose an extended hierarchical Kriging (EHK) method that efficiently accommodates non-level datasets, extending the existing body of research in this domain. The EHK method introduces innovative techniques to decrease the size of the correlation matrix and minimize the number of scaling factors, independent of the quantity of LF models involved. As a result, it substantially reduces the computational expenses associated with model processing while upholding superior accuracy levels in comparison to established methods.

In that regard, this work contributes to the progress of the research field in the following aspects.

- This work proposes an EHK framework to incorporate multiple non-level LF models that are taken as the trend functions and assembled by allocating different scaling factors. Subsequently, a Gaussian process model is employed to formulate the discrepancy function, representing the variance between the HF model and the ensembled LF models.

- In the model-processing phase, a strategy is employed so that the implicit estimation of scaling factors occurs concurrently with the optimization of other hyperparameters within the discrepancy model. Essentially, these scaling factors are formulated as functions of the discrepancy model's hyperparameters. This approach effectively diminishes the number of independent parameters subject to optimization, irrespective of the quantity of LF models integrated.

- Furthermore, it is worth noting that the computational complexity of the EHK algorithm is inferior to that of other rivals. Multiple investigations carried out in this study substantiate that the EHK model offers reduced computational costs compared to other competing methods, all the while meeting the necessary level of accuracy standards.

  Through numerical analysis, the following findings can be drawn.

- The proposed EHK method offers enhanced capabilities for constructing more effective approximation models, particularly for multi-fidelity and high-dimensional datasets, in comparison to traditional methodologies. One notable advantage is the reduction in the size of the correlation matrix within EHK, leading to decreased algorithmic complexity and, consequently, significantly reduced computational processing time.

- In the EHK method, the optimization of scaling factors, which encapsulate the influence of LF models on HF model predictions, is implicitly performed during model processing. As a consequence, EHK model construction exhibits substantially lower computational costs while still maintaining the requisite level of accuracy. This advantage becomes particularly pronounced when a considerable number of LF models are integrated into the framework employed for constructing the HF model.

The structure of this paper is organized as follows: Section 2 presents an extensive literature review, offering insights into related works within the same research domain. Section 3 provides a succinct exposition of both the original HK method and the proposed EHK method, rooted in the comprehensive HK framework. Section 4 is dedicated to the evaluation of the EHK method, encompassing a spectrum of multi-dimensional analytical examples and two practical engineering case studies. These particular case studies pertain to the construction of an approximation model for the fluidized bed process and for an aerodynamic database of an electric vertical take-off and landing (eVTOL) vehicle designed for urban air mobility, denoted as KP-2, under diverse flight conditions. Finally, Section 5 encapsulates this research with a conclusive summary, highlighting the technical contributions and significant insights derived from this research endeavor.

## 2. Related Work

Prior publications have proposed multi-level MFSM frameworks for incorporating non-level data based on the weighted average of statistical surrogate models [34–38]. Several LF models were created in this scenario by utilizing various methods to simplify the HF model, leading to non-level LF models with varying degrees of correlation to the HF model in the subregion of the design space. Parallelly, Chen et al. [39] proposed three non-hierarchical multi-model fusion approaches based on spatial-random-process, denoted as WS, PC-DIT, PC-CSC, to incorporate multiple non-level LF datasets with an HF dataset in different Bayesian frameworks. Zhang et al. [37] proposed a linear-regression-based multi-fidelity surrogate model (LRMFS) that can incorporate multiple non-level LF models into the HF model at a low computational cost. M. Xiang et al. [36] proposed extended co-Kriging (ECK) to incorporate non-level LF models to improve the HF model's accuracy. Zhang et al. [38] proposed an improved version of ECK, called NHLF-co-Kriging, featuring a different strategy to obtain optimal scaling factors of the LF models.

A summary of related works in the literature, previously mentioned in the article's introduction, including (i) characteristics of low-fidelity levels, and (ii) types of MFSM frameworks, is shown in Table 1.

**Table 1.** Summary of multi-level MFSM methods.

| Methods | Granularity of LF Datasets | | MFSM Framework Type |
|---|---|---|---|
| | **Multi-Level** | **Non-Level** | |
| Co-Kriging, 2000 [22,24,30] | √ | | Bayesian |
| HK, 2012 [25] | √ | | Bayesian |
| MCK, 2012 [24] | √ | | Bayesian |
| IHK, 2018 [12] | √ | | Bayesian |
| MVCM, 2015 [19] | √ | | BF |
| GVFM, 2015 [20] | √ | | BF |
| co-BRF, 2017 [40] | √ | | Bayesian |
| MFGP, 2018 [41] | √ | | Bayesian |
| POD-co-Kriging, 2018 [42] | √ | | Bayesian |
| SM-VFM, 2018 [26] | √ | | SM, BF |
| MHK, 2020 [31,32] | √ | | Bayesian |
| MDNN, 2020 [13] | √ | | BF |
| GCK, 2020 [43,44] | √ | | Bayesian |
| TL-VFSM, 2021 [27] | √ | | BF |
| GAN-MDF, 2022 [29] | √ | | BF |
| MMGP, 2021 [45] | √ | | Bayesian |
| WS, PC-DIT, PC-CSC, 2016 [39] | | √ | Bayesian |
| ECK, 2018 [36] | | √ | Bayesian |
| LRMFS, 2018 [37] | | √ | BF |
| VWS-MFS, 2021 [46] | | √ | Bayesian |
| NHLF-co-Kriging, 2022 [38] | | √ | Bayesian |
| Our work (EHK) | | √ | Bayesian |

Based on the previous summary, most existing methods utilize Bayesian discrepancy frameworks to construct the HF model based on prior information from LF models. However, these methods often require an increasingly high computational cost with a large number of LF datasets incorporated due to the increase in the number of hyperparameters to be estimated. Furthermore, the cost of model processing increases exponentially with sample size, dimensionality, and the number of LF models, primarily due to the complex calculations involved in inverting the correlation matrix and tuning a large number of hyperparameters. Although the LRMFS method addresses this issue by using a low-order polynomial to construct the discrepancy model and solving sets of linear equations to obtain scaling factors and model coefficients quickly, it still suffers from limitations related to matrix inversion and data with intricate landscapes.

## 3. Proposed Methodology

### 3.1. Preliminaries

In this section, the comparison between existing multi-level MFSM methods will be discussed to evaluate the pros and cons of each approach for dealing with data fusion problems featuring multiple non-level LF datasets. For an m-dimensional problem, the sampling plan of the HF datasets, $x_{HF}$, and the corresponding responses, $y_{HF}$, are denoted by

$$x_{HF} = \left\{ x_{HF}^{(1)}, x_{HF}^{(2)}, \dots, x_{HF}^{(n_{HF})} \right\}^{\mathrm{T}} \tag{1}$$

$$y_{HF} = \left\{ y_{HF}\left(x_{HF}^{(1)}\right), y_{HF}\left(x_{HF}^{(2)}\right), \dots, y_{HF}\left(x_{HF}^{(n_{HF})}\right) \right\}^{\mathrm{T}} \tag{2}$$

The sampling plans and the corresponding observations of the $L$ levels of LF datasets are denoted by

$$x_{LF,i} = \left\{ x_{LF,i}^{(1)}, x_{LF,i}^{(2)}, \dots, x_{LF,i}^{(n_{LF,i})} \right\}^{\mathrm{T}}, i = \overline{1, L} \tag{3}$$

$$y_{LF,i} = \left\{ f_{LF,i}\left(x_{LF,i}^{(1)}\right), f_{LF,i}\left(x_{LF,i}^{(2)}\right), \dots, f_{LF,i}\left(x_{LF,i}^{(n_{LF,i})}\right) \right\}^{\mathrm{T}}, i = \overline{1, L} \tag{4}$$

where $n_{HF}$ and $n_{LF,i}, i = \overline{1, L}$ are the number of samples in HF and LF datasets. It is usually assumed that $n_{LF,i}$ is large enough to construct the correct LF models ($n_{LF,i} \gg n_{HF}, i = \overline{1, L}$).

Conventional multi-level MFSM methods assume that LF datasets can be easily distinguished based on their fidelity levels across the design space. In these methods, the ultimate HF model is constructed recursively, typically involving $L$ stages where LF models are sequentially built from LF datasets until the final HF model is obtained. The information from LF models is transmitted sequentially between adjacent fidelity levels, and the accuracy of the HF model are directly influenced by the $L$th-level model, $\hat{f}_{LF,L}(x)$. One example of such an approach is the HK modeling method proposed by Han and Gortz [31,32]. However, these recursive MFSM frameworks are not suitable for scenarios involving multiple non-level LF datasets, where the importance of LF datasets is similar or the fidelity levels between LF datasets are challenging to identify. Furthermore, the influence of LF models on the ultimate HF model primarily relies on the LF model in $L$th-level, $\hat{f}_{LF,L}(x)$, limiting the efficient incorporation of information from other LF models.

In contrast, the ECK method proposed by Xiao et al. [36] offers a more reasonable approach in such scenarios by employing a non-recursive framework and combining non-level LF models using scaling factors. This approach directly enhances the accuracy of the HF model by incorporating information from all LF datasets. However, the scalability of the ECK method becomes challenging when incorporating a large number of LF datasets since the estimation of numerous scaling factors increases computational complexity and modeling costs.

To address these challenges, the proposed EHK method adopts a non-recursive framework while introducing a strategy to derive unknown scaling factors from the correlation hyperparameters of the discrepancy mode. This approach reduces the number of tun-

ing hyperparameters required in the construction of the discrepancy model, resulting in decreased computational costs and modeling complexity.

Multi-Level Hierarchical Kriging Model

The two-level HK method operates under the assumption that sample points can be categorized into two levels: HF and LF datasets. The HF sample points are derived from expensive methods and provide highly accurate information. On the other hand, the LF sample points are obtained from less computationally demanding methods, resulting in lower accuracy but higher accessibility in terms of quantity. In this approach, the LF model is utilized to capture the global characteristics of the HF model, while the HF samples are employed to correct any errors in the model. The HK framework can be written in the form:

$$\hat{f}_{HF}(x) = \rho_0 \hat{f}_{LF}(x) + Z_d(x) \tag{5}$$

Here, $\hat{f}_{LF}(x)$ is a Kriging model of LF samples. $\rho_0$ is an unknown constant scaling factor indicating the influence of the LF model on the behavior of the HF model. $Z_d(x)$ is a stationary random process having zero mean and a covariance $\sigma_d^2$, denoting the discrepancy model between $\hat{f}_{HF}(x)$ and $\rho \hat{f}_{LF}(x)$.

$$Cov\left[Z(x), Z(x')\right] = \sigma_d^2 \mathbf{R}(x, x') \tag{6}$$

where $\mathbf{R}(x, x')$ is the spatial correlation matrix of observed data. The HK predictor at an untried point $x$ has the form as:

$$\hat{f}_{HF}(x) = \rho_0 \hat{f}_{LF}(x) + \mathbf{r}(x)^{\mathrm{T}} \mathbf{R}^{-1} \left( \mathbf{y}_{HF} - \rho_0 \widehat{\mathbf{F}}_{LF} \right) \tag{7}$$

where $\rho_0 = \left( \widehat{\mathbf{F}}_{LF}^{\mathrm{T}} \mathbf{R}^{-1} \widehat{\mathbf{F}}_{LF} \right)^{-1} \widehat{\mathbf{F}}_{LF}^{\mathrm{T}} \mathbf{R}^{-1} \mathbf{y}_{HF}$ and $\widehat{\mathbf{F}}_{LF} = \left[ \hat{f}_{LF}\left( x_{HF}^{(1)} \right), \ldots, \hat{f}_{LF}\left( x_{HF}^{(n_{HF})} \right) \right]$ are the vectors of estimated responses of the LF model at the training HF sample points. $\mathbf{y}_{HF}$ is the vector of the HF samples' responses. $\mathbf{r}(x) = \mathbf{R}(x, \mathbf{x}_{HF})$ is a vector of the estimated correlation between the untried point $x$ and the HF training samples $\mathbf{x}_{HF}$.

The HK model for multiple hierarchical levels of fidelity is defined in a recursive manner as [31,32]:

$$\hat{f}_i(x) = \begin{cases} \rho_{i-1} + Z_{i-1}(x) & i = 1 \\ \rho_{i-1} \hat{f}_{i-1}(x) + Z_{i-1}(x) & i = 2, \ldots, L, L+1, \end{cases} \tag{8}$$

Here, $L$ is the number of LF simulation models with varying degrees of fidelity and computational expense. $i$ denotes the $i$th level of fidelity, with "$i = 1$" and "$i = L + 1$" representing the lowest and highest and fidelity levels, respectively.

### 3.2. The Proposed EHK Method

The EHK framework integrates the data from an HF dataset and $L$ non-level LF datasets. The formula of the EHK model for the approximation of the HF model can be written as

$$\hat{f}_{HF}(x) = \sum_{i=1}^{L} \rho_i \hat{f}_{LF,i}(x) + Z_d(x) \tag{9}$$

The EHK model can also be written in an alternative form as

$$\hat{f}_{HF}(x) = \boldsymbol{\rho}^{\mathrm{T}} \widehat{\boldsymbol{f}}_{LF}(x) + Z_d(x) \tag{10}$$

where $\widehat{\boldsymbol{f}}_{LF}(x) = \left[ \hat{f}_{LF,1}(x), \ldots, \hat{f}_{LF,L}(x) \right]^{\mathrm{T}}$ is a vector of non-level LF models, which can be built directly by a Kriging method or other approximation methods with LF datasets. The LF models are scaled by the unknown constant scaling factors $\boldsymbol{\rho} = [\rho_1, \rho_2, \ldots, \rho_L]^{\mathrm{T}}$, serving as the global trend model. $Z_d(x)$ is a stationary random process having zero mean and a

covariance $\sigma_d^2$, denoting the discrepancy model between the HF model and the ensembled LF models. Hence, the prediction of the EHK model at any untried $x$ can be obtained:

$$\hat{f}_{HF}(x) = \boldsymbol{\rho}^{\mathbf{T}}\widehat{\boldsymbol{f}}_{\boldsymbol{LF}}(x) + \boldsymbol{r}(x)^{\mathbf{T}}\boldsymbol{R}^{-1}\left(\boldsymbol{y}_{HF} - \boldsymbol{\rho}^{\mathbf{T}}\widehat{\boldsymbol{F}}_{LF}\right) \tag{11}$$

where $\widehat{\boldsymbol{F}}_{LF} = \left[\widehat{\boldsymbol{f}}_{LF}\left(x_{HF}^{(1)}\right),\ldots\widehat{\boldsymbol{f}}_{LF}\left(x_{HF}^{(n_{HF})}\right)\right] \in \mathbb{R}^{L \times n_{HF}}$ is the matrix of the estimated responses of LF models at HF samples, $\boldsymbol{R}(x_{HF}, x'_{HF})$ is the correlation matrix between HF samples, Equation (12), and $\boldsymbol{r}(x) = \phi(x, x_{HF})$ is a vector of the estimated correlation between the untried point $x$ and the HF training samples. $\phi(x, x')$ is the spatial correlation function, which only depends on the Euclidean distance between two sides $x$ and $x'$. Compared to the previous original publication, this article adopted the well-known Gaussian exponential function with second order, as shown in Equation (13), to reduce the computational complexity of model construction while still maintaining the critical features of the HK model in the proposed EHK model. $\boldsymbol{\theta} = [\theta_1, \theta_2, \ldots, \theta_m] \in \mathbb{R}^m$ is a vector of unknown hyperparameters in the Gaussian exponential function.

$$\boldsymbol{R}\left(x_{HF}, x'_{HF}\right) = \left(\phi\left(x_{HF}, x'_{HF}\right)\right)_{i,j} \in \mathbb{R}^{n_{HF} \times n_{HF}} \tag{12}$$

$$\phi\left(x, x'\right) = \prod_{k=1}^{m} \exp\left(-\theta_k \left|x^{(k)} - x'^{(k)}\right|^2\right) \tag{13}$$

The predicted error of the EHK model for an untried point can be written as

$$\hat{s}^2(x) = \sigma^2\left\{1 - \boldsymbol{r}^{\mathbf{T}}\boldsymbol{R}^{-1}\boldsymbol{r} + \left(\widehat{\boldsymbol{F}}_{LF}{}^{\mathbf{T}}\boldsymbol{R}^{-1}\boldsymbol{r}(x) - \widehat{\boldsymbol{f}}_{LF}(x)\right)^{\mathbf{T}}\left(\widehat{\boldsymbol{F}}_{LF}{}^{\mathbf{T}}\boldsymbol{R}^{-1}\widehat{\boldsymbol{F}}_{lf}\right)^{-1}\left(\widehat{\boldsymbol{F}}_{LF}{}^{\mathbf{T}}\boldsymbol{R}^{-1}\boldsymbol{r}(x) - \widehat{\boldsymbol{f}}_{LF}(x)\right)\right\} \tag{14}$$

To estimate the hyperparameters $\boldsymbol{\rho}$, $\boldsymbol{\theta}$, and $\sigma_d^2$, the maximum likelihood estimation (MLE) method is used to maximize the likelihood function given by

$$L\left(\boldsymbol{\rho}, \sigma_d^2, \boldsymbol{\theta}\right) = \frac{1}{\sqrt{\left(2\pi\sigma_d^2\right)^{n_{HF}}|\boldsymbol{R}|}}\exp\left(-\frac{1}{2}\frac{\left(\boldsymbol{y}_{HF} - \boldsymbol{\rho}^{\mathbf{T}}\widehat{\boldsymbol{F}}_{LF}\right)^{\mathbf{T}}\boldsymbol{R}^{-1}\left(\boldsymbol{y}_{HF} - \boldsymbol{\rho}^{\mathbf{T}}\widehat{\boldsymbol{F}}_{LF}\right)}{\sigma_d^2}\right) \tag{15}$$

Taking the natural logarithm of the likelihood function, the simplified form is achieved for being maximized.

$$\ln\{L(\theta)\} = -n_{HF}\ln\left(\sigma_d^2\right) - \ln|\boldsymbol{R}| - \frac{\left(\boldsymbol{y}_{HF} - \boldsymbol{\rho}^{\mathbf{T}}\widehat{\boldsymbol{F}}_{LF}\right)^{\mathbf{T}}\boldsymbol{R}^{-1}\left(\boldsymbol{y}_{HF} - \boldsymbol{\rho}^{\mathbf{T}}\widehat{\boldsymbol{F}}_{LF}\right)}{\sigma_d^2} \tag{16}$$

The derivatives with respect to $\boldsymbol{\rho}$ and $\sigma^2$ are set to zero. The MLEs of the unknown scaling factors and the process variances are analytically calculated as

$$\boldsymbol{\rho}^*(\boldsymbol{\theta}) = \left(\widehat{\boldsymbol{F}}_{LF}{}^{\mathbf{T}}\boldsymbol{R}^{-1}\widehat{\boldsymbol{F}}_{LF}\right)^{-1}\widehat{\boldsymbol{F}}_{LF}{}^{\mathbf{T}}\boldsymbol{R}^{-1}\boldsymbol{y}_{HF} \tag{17}$$

$$\sigma_d^2(\boldsymbol{\theta}) = \frac{1}{n_{HF}}\left(\boldsymbol{y}_{HF} - \boldsymbol{\rho}^{\mathbf{T}}\widehat{\boldsymbol{F}}_{LF}\right)^{\mathbf{T}}\boldsymbol{R}^{-1}\left(\boldsymbol{y}_{HF} - \boldsymbol{\rho}^{\mathbf{T}}\widehat{\boldsymbol{F}}_{LF}\right) \tag{18}$$

Here, the vector of scaling factors is $\boldsymbol{\rho}$, which is transformed into a function of hyperparameters $\boldsymbol{\theta}$. Substituting Equations (17) and (18) into Equation (16), the following expression is left to be maximized:

$$\ln\{L(\theta)\} = -n_{HF}\ln\left(\sigma_d^2\right) - \ln|\boldsymbol{R}| - \frac{\left(\boldsymbol{y}_{HF} - \boldsymbol{\rho}^{\mathbf{T}}\widehat{\boldsymbol{F}}_{LF}\right)^{\mathbf{T}}\boldsymbol{R}^{-1}\left(\boldsymbol{y}_{HF} - \boldsymbol{\rho}^{\mathbf{T}}\widehat{\boldsymbol{F}}_{LF}\right)^{\mathbf{T}}}{\sigma_d^2} \tag{19}$$

The unknown hyperparameters $\theta$ are found by maximizing Equation (19). Since there is no closed-form solution for $\theta$, it has to be found by numerical optimization, which has the form.

$$\theta = \text{argmax}[\ln\{L(\theta)\}] \tag{20}$$

Note that the scaling factors $\rho = [\rho_1, \rho_2, \ldots, \rho_L]$ are implicitly tuned during tuning of the hyperparameters. Thus, it is worth noting that the optimization problem associated with model processing in the EHK method consistently features a variable count equivalent to the number of dimensions, irrespective of the number of incorporated LF models. Consequently, the EHK method exhibits a more concise set of hyperparameters compared to the ECK model [36]. In this study, the *genetic algorithm* (GA) function from the Global Optimization Toolbox in MATLAB was employed to tackle the optimization problems related to hyperparameters and avoid the problem of local optimum solutions. Additionally, the range of hyperparameters, $\theta$, was constrained within the bounds of $[10^{-2}, 10^3]$. In all testing cases within this study, the GA solver was implemented with a population size of 200 and a maximum number of generations equal to $100 \times n_{hyp}$, where $n_{hyp}$ presents the total count of tuning hyperparameters. A comprehensive algorithm for the model-processing procedure is elaborated with a flowchart and a pseudocode, as delineated in Figure 1 and Algorithm 1, respectively.

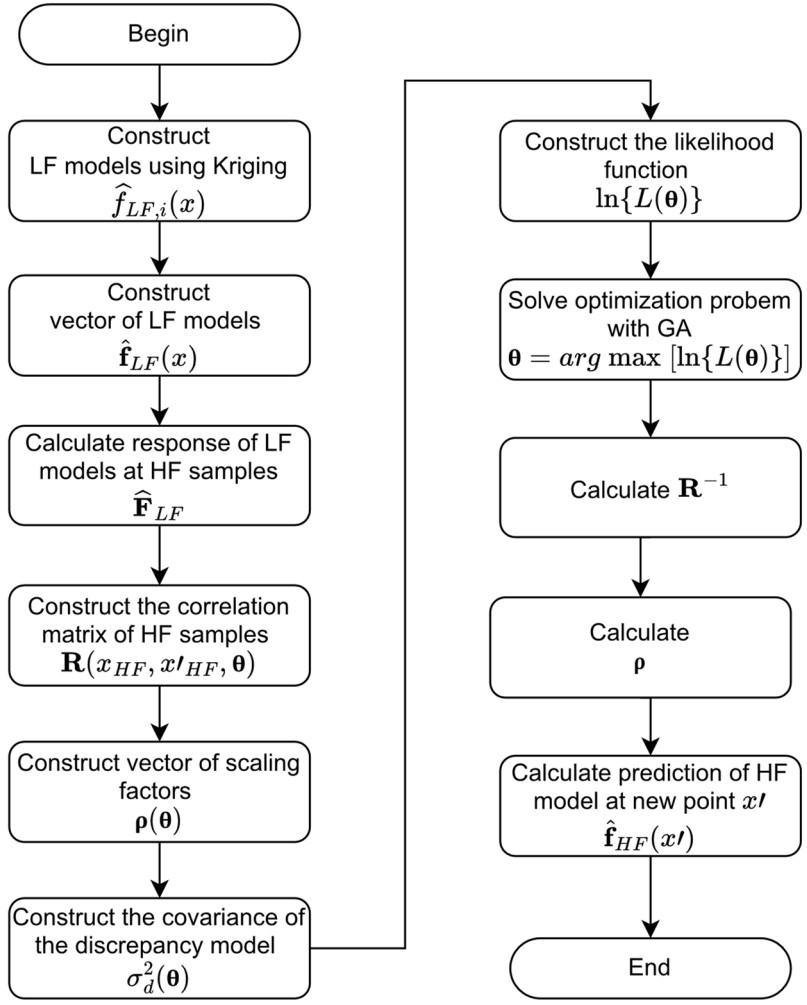

**Figure 1.** The flowchart of EHK model construction.

**Algorithm 1:** Algorithm of EHK model construction

**Input**: The LF sampling datasets $S_{LF_i} = \left\{ x_{LF,i}, y_{LF,i} \right\}, i = \overline{1, L}$ and HF sampling data $S_{HF} = \{ x_{HF}, y_{HF} \}$.
**Output**: the predictor of the EHK model $\hat{f}_{HF}(x)$.

1: **Begin**
2:    $\hat{f}_{LF,i}(x) \leftarrow$ Construct LF models from LF datasets using Kriging.
3:    $\widehat{f}_{LF}(x) \leftarrow$ Construct vector of LF models
4:    $\widehat{F}_{LF} \leftarrow$ Calculate the matrix of the estimated responses of LF models at HF samples.
5:    $R(x_{HF}, x'_{HF}, \theta) \leftarrow$ Construct the correlation matrix of HF samples containing unknown hyperparameters $\theta$, as in Equation (12).
6:    $\rho(\theta) \leftarrow$ Construct the vector of the scaling factors as a function of unknown hyperparameters $\theta$, as in Equation (17).
7:    $\sigma_d^2(\theta) \leftarrow$ Construct the covariance of the discrepancy model as a function of unknown hyperparameters $\theta$, as in Equation (18).
8:    $\ln\{ L(\theta) \} \leftarrow$ Construct the logarithm form of the likelihood function as a function of unknown hyperparameters $\theta$, as Equation (19).
9:    $\theta \leftarrow$ Solve Equation (20) for the optimal solution of $\theta$ using GA solver.
10:   $R^{-1} \leftarrow$ Calculate inverse matrix of the correlation matrix from resulting $\theta$.
11:   $\rho \leftarrow$ Calculate the vector of the scaling factors from resulting $\theta$.
12:   $\widehat{f}_{HF}(x') \leftarrow$ Calculate the prediction of the EHK model at the new point $x'$, using Equation (11)
13: **End.**

### 3.3. Computational Complexity and Cost Analysis

While it is feasible to enhance the predictive capacity of the proposed EHK model through the integration of numerous non-level LF models or by augmenting the HF sample size, the computational complexity of the model processing is fundamentally contingent on the inversion of the correlation matrix, denoted as $R^{-1}$. Furthermore, the challenges associated with dimensionality, large dataset sizes, and a multitude of LF models render the estimation of hyperparameters a complex high-dimensional optimization endeavor. The computational expenses incurred during the process of hyperparameter estimation stem from the following key factors:

1. The number of training samples: As the quantity of training samples rises, it invariably augments the size of the correlation matrix, a fundamental component in many Gaussian-process-based modeling methodologies. Consequently, this amplifies the computational complexity required for the inversion of the correlation matrix. The consequence is a substantial demand for computational resources and an increase in computational time, resulting in heightened costs.

2. The number of tuning hyperparameters: Various MFSM methods necessitate distinct quantities of tuning hyperparameters due to their particular model formulations. Typically, these hyperparameters encompass:

   - Correlation parameters: The quantity of correlation parameters is contingent upon the chosen correlation function inherent to each MFSM method. In the case of the proposed EHK model, the Gaussian exponential function is employed, resulting in the number of correlation parameters equating to the number of dimensions within the dataset.

   - Scaling factor: The number of scaling factors corresponds to the number of integrated LF datasets.

   - Other parameters: Some methodologies incorporate supplementary parameters into their models.

3. The optimization algorithm: In this study, the GA is utilized to optimize hyperparameters, which is especially effective for high-dimensional problems. The computational cost associated with running a GA, typically quantified in terms of computation time or the number of function evaluations, notably escalates as the number of tuning hyperparameters increases.

To mitigate the impact of programming techniques and computational resources, the computational cost is measured using the number of function evaluations (FEs) of the likelihood function, which serves as an efficient metric during model processing. The cost of evaluating the likelihood function is primarily associated with the inversion of the correlation matrix. Taking these factors into account, it becomes evident that increasing the number of integrated datasets, data dimensions, and size leads to a greater number of tuning hyperparameters, consequently escalating the FEs as well as the computational cost when utilizing GA.

Table 2 lists the computational complexity of calculating $\boldsymbol{R}^{-1}$ for each three investigated MFSM methods in the second column; additionally, the last two columns provide the number of independent hyperparameters, including $n_s$ scaling factors and $n_c$ correlation parameters. Hence, the total number of hyperparameters is $N_{hyp} = n_s + n_c$. For general single-output multi-fidelity models, the computational complexity is $O(n^3)$ [47]. In the case of the EHK model, it requires calculating the inverse of an $n_{HF} \times n_{HF}$ correlation matrix (Equation (12)) leading to a computational complexity of $O(n_{HF}^3)$. Only the correlation parameters $\boldsymbol{\theta} = [\theta_1, \theta_2, \ldots, \theta_m] \in \mathbb{R}^m$ are the independent hyperparameters in the EHK model and the scaling factors can be derived from the correlation parameters. On the other hand, the ECK, WS, PC-DIT, and PC-CSC models require calculating the inverse of an $N \times N$ $\left(N = \sum_1^L n_{LF,i} + n_{HF}\right)$ correlation matrix, leading to a computational complexity of $O(N^3)$. Additionally, all scaling factors $\rho$ and correlation parameters $\theta$ are adopted as the independent hyperparameters in the ECK model. For the LRMFS model, the model processing required calculating the inverse of the augmented design matrix [37], which has a size depending on the number of scaling factors and polynomial coefficients.

**Table 2.** The characteristics of different models, including the computational complexity of $R^{-1}$ and the number of hyperparameters to be inferred.

| Model | $R^{-1}$ | $n_s$ | $n_c$ |
|---|---|---|---|
| EHK | $O(n_{HF}^3)$ | 0 | $m$ |
| EHK | $O(n_{HF}^3)$ | 0 | $m$ |
| ECK | $O(N^3)$ | $L$ | $(L+1) \times m$ |
| LRMFS | $O\left((L+p)^3\right)$ | $L$ | $p$ |
| WS | $O(N^3)$ | $L$ | $(L+1) \times m + 1$ |
| PC-DIT | $O(N^3)$ | 0 | $(L+1) \times m + 1$ |
| PC-CSC | $O(N^3)$ | 0 | $(L+1) \times m + 1$ |

In summary, the computational complexity of the EHK model is markedly reduced in comparison to the ECK, WS, PC-DIT, and PC-CSC models, primarily attributed to the EHK model's utilization of a smaller-sized correlation matrix and fewer tuning hyperparameters. Additionally, the EHK model maintains an advantage by requiring hyperparameters solely for the correlation basis function, irrespective of the number of LF models incorporated. Consequently, the EHK method features a smaller number of tuning hyperparameters in comparison to other referenced techniques, thereby resulting in a reduced computational cost for hyperparameter estimation.

In contrast to Bayesian models, LRMFS model parameters can be estimated without the need for an optimization process, which significantly reduces computational costs, particularly in low-dimensional problems. However, it is important to note that in high-dimensional data cases, the number of polynomial coefficients, $p$, may increase substantially, resulting in a significant rise in the computational complexity of the LRMFS model.

Due to the above reasons, the proposed EHK modeling method is more appropriate for incorporating multiple non-level LF datasets compared with the existing multi-level MFSM methods.

## 4. Numerical Settings and Experiments

In this section, several multi-dimensional analytical examples were used to validate the proposed EHK model. Furthermore, an engineering case study of modeling an aerodynamic database of high-speed aircraft was used to illustrate the merits and effectiveness of the proposed EHK model, by comparing it with conventional approaches using Kriging, ECK, HK, and LRMFS. The relative root mean square error (RRMSE) and relative maximum absolute error (RRMAE) were adopted to validate the surrogate model's global and local accuracy, respectively. The smaller the RRMSE and RMAE, the more accurate the model is. The expressions of these two metrics [12] are

$$RRMSE = \frac{1}{STD}\sqrt{\frac{1}{n_{test}}\sum_{i=1}^{n_{test}}\left(y_i - \hat{f}_i\right)^2} \tag{21}$$

$$RMAE = \frac{1}{STD}\max\left|y_i - \hat{f}_i\right|, \ i \in \overline{1, n_{test}} \tag{22}$$

$$STD = \sqrt{\frac{1}{n_{test}}\sum_{1}^{n_{test}}(y_i - \overline{y}_i)^2} \tag{23}$$

where $n_{test}$ is the total number of testing points, $\hat{f}_i$ is the predicted response of the testing points, $y_i$ is the true responses of the testing points, and $\overline{y}_i$ and STD are the mean and standard deviation of all testing points, respectively.

### 4.1. One-Dimensional Example

In this work, a one-dimensional numerical example is used to test the approximation capability of the EHK model. In this example, the analytical function in Equation (24) represents the HF function, and two LF functions [39] are given in Equations (25) and (26).

$$f_{HF}(x) = \sin x \tag{24}$$

$$f_{LF_1}(x) = \sin x + 0.1(x - \pi)^2 \tag{25}$$

$$f_{LF_2}(x) = 1.2\sin x + 0.1(x - \pi)^2 - 0.2 \tag{26}$$

Using the Latin hypercube sampling (LHS) method, two LF sampling plans were generated, denoted as $x_{LF_1}$ and $x_{LF_2}$, along with one HF sampling plan, $x_{HF}$, as shown in Table 3. A demonstration of various testing models and sampling sets is shown in Figure 2. It can be observed that the two LF models only partially captured the trends exhibited by the HF model within specific localized regions of the design space. On a global scale, determining which LF model better represented the correct trends of the HF model posed a challenge. Hence, in this scenario, the fidelity levels of the LF models were indistinguishable, classifying them as non-level LF models.

**Table 3.** The *x* locations of the HF and LF sample plans [39].

| Datasets | Sample Points |
|---|---|
| $x_{HF}$ | $\{1.0226, 2.2300, 5.5210\}^{\mathrm{T}}$ |
| $x_{LF_1}$ | $\{3.6236, 1.928, 1.853, 2.6127, 4.578, 0.4317, 0.7170, 5.9766, 5.4798, 4.3535\}^{\mathrm{T}}$ |
| $x_{LF_2}$ | $\{3.6236, 1.928, 1.853, 2.6127, 4.578, 0.4317, 0.7170, 5.9766, 5.4798, 4.3535\}^{\mathrm{T}}$ |

Next, all multi-fidelity training datasets were used to generate different approximation models using the EHK, ECK, LRMFS, and MHK methods. Table 4 shows the assignments of training datasets for different models. In this stage, the EHK, ECK, and LRMFS models were constructed using the HF dataset $S_{HF}$ and two LF datasets $S_{LF_1}$ and $S_{LF_2}$ without identifying the fidelity levels between LF datasets. Additionally, two models MHK1 and MHK2 were constructed using the MHK method with all datasets, $S_{HF}$, $S_{LF_1}$ and $S_{LF_2}$. In

this example, the fidelity levels between training datasets in the MHK1 and MHK2 models were distinctly identified. In the MHK1 model, $S_{HF}$ was labeled as the HF data, $S_{LF_1}$ was labeled as the middle-fidelity (MF) data, and $S_{LF_2}$ was labeled as the LF data. In contrast, $S_{LF_1}$ was labeled as the LF data and $S_{LF_2}$ was labeled as the MF data in the MHK2 model. Both MHK1 and MHK2 aimed to construct the final model of the HF data in a recursive manner, as shown in Equation (8).

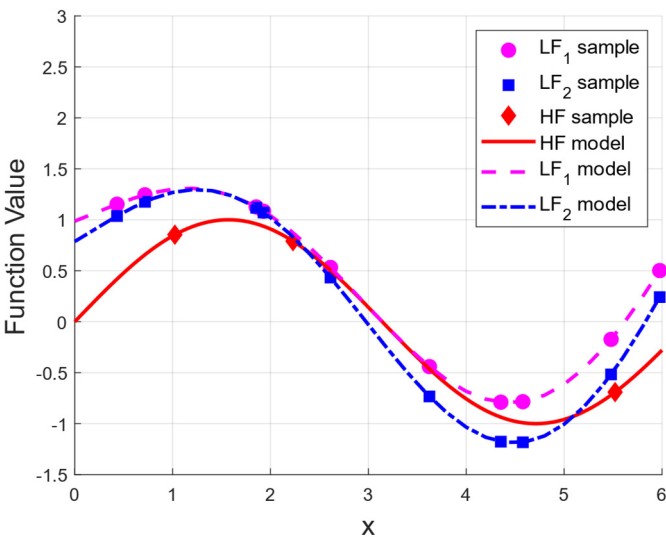

**Figure 2.** One dimensional testing models and sets of sample points.

**Table 4.** Training datasets in different models.

| Dataset | EHK | ECK | LRMFS | MHK1 | MHK2 | Kriging |
|---|---|---|---|---|---|---|
| $S_{HF} = \{\mathbf{x}_{HF}, \mathbf{y}_{HF}\}$ | $\sqrt{}$ (HF) | $\sqrt{}$ (HF) | $\sqrt{}$ (HF) | $\sqrt{}$ (HF) | $\sqrt{}$ (HF) | $\sqrt{}$ (HF) |
| $S_{LF_1} = \{\mathbf{x}_{LF_1}, \mathbf{y}_{LF_1}\}$ | $\sqrt{}$ (LF) | $\sqrt{}$ (LF) | $\sqrt{}$ (LF) | $\sqrt{}$ (MF) | $\sqrt{}$ (LF) | --- |
| $S_{LF_2} = \{\mathbf{x}_{LF_2}, \mathbf{y}_{LF_2}\}$ | $\sqrt{}$ (LF) | $\sqrt{}$ (LF) | $\sqrt{}$ (LF) | $\sqrt{}$ (LF) | $\sqrt{}$ (MF) | --- |

Figure 3a presents a comparative analysis of the constructed models with respect to the prediction of the HF model. In this instance, an additional step was taken to conduct extrapolations within limited regions of [0.0, 1.0] and [5.5, 6.5]. An HF validation dataset (red diamonds) was employed to validate the performance of the resultant models. The pointwise errors between the resulting models and the validation data are illustrated in Figure 3b,c. Given the minuscule pointwise errors of the EHK and ECK models, which are challenging to discern in comparison to other models in the same figure, Figure 3c represents the pointwise errors of the resulting models using a logarithmic scale on the y-axis. The results unequivocally demonstrate the superior performance of the EHK model in capturing the HF function with the lowest pointwise error. Conversely, the LRMFS model exhibited the poorest performance, failing to accurately capture the tendency of the HF data. Additionally, the LRMFS model did not accurately represent the HF input sample at x = 2.23, leading to a substantial error at x = 2.23, as depicted in Figure 3b. This occurrence signified an underfitting phenomenon within the LRMFS model, a result of the limited number of training HF samples and the occurrence of a singular matrix issue during model construction [37], leading to a reduction in the LRMFS model's accuracy compared to other models.

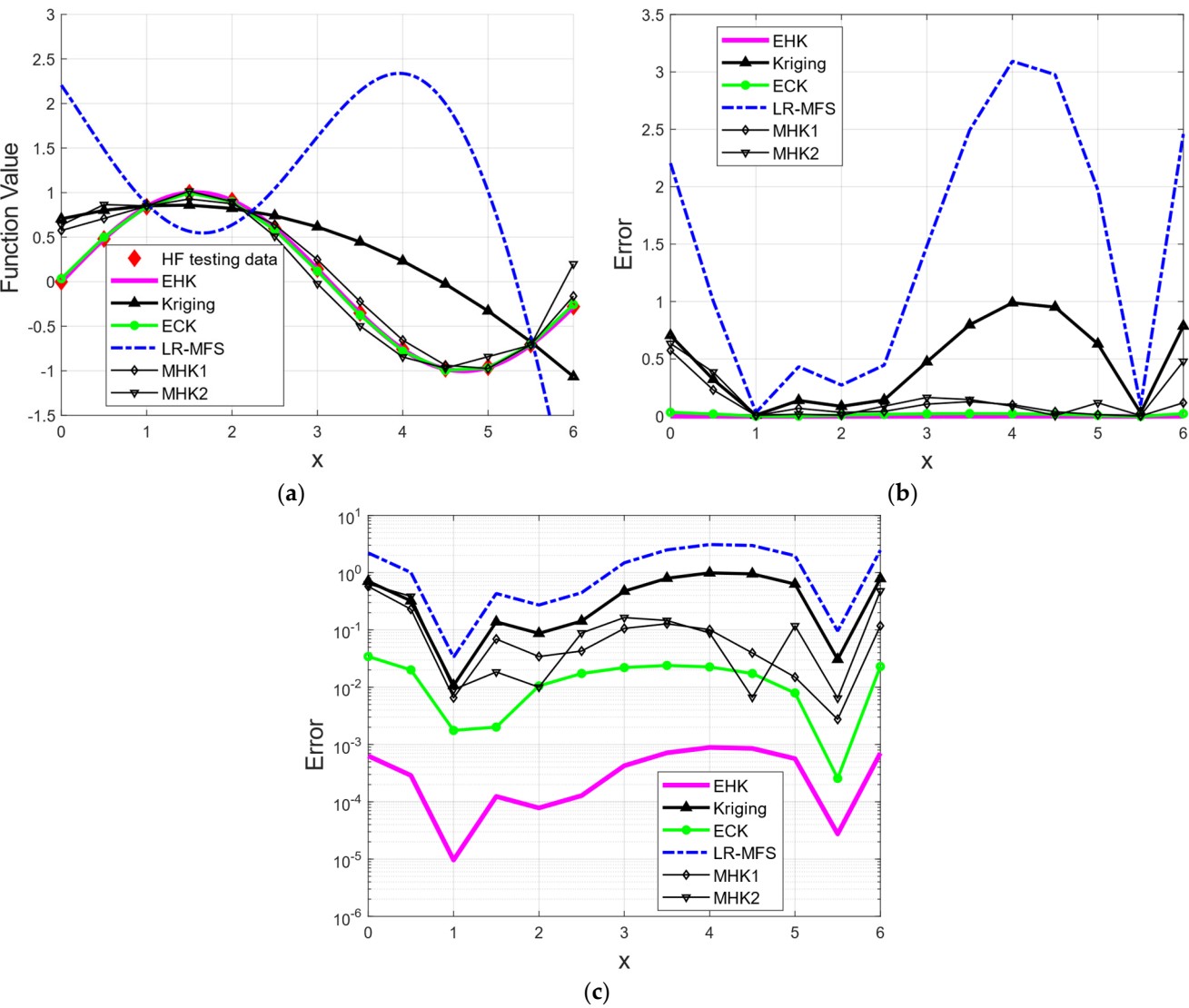

**Figure 3.** Comparison of different surrogate models: (**a**) prediction models; (**b**) pointwise error in normal scale; (**c**) pointwise error in base ten logarithmic scale on the y axis.

It is imperative to emphasize that the MHK1 and MHK2 models depended on the tendencies exhibited by the MF datasets to function as trend models for capturing the global behavior of the HF model. However, they failed to optimally exploit the information made available by the LF datasets at their disposal. Consequently, the construction of the HF model became heavily contingent on the necessity for accurate and subjectively labeled fidelity levels between LF datasets. In contrast, the EHK model adeptly integrated and concurrently fused information derived from both LF models, facilitating the creation of a substantially more robust trend model that effectively encapsulated the behavior of the HF model. This proficiency led to the EHK model achieving higher accuracy in comparison to the MHK1 and MHK2 models.

Table 5 shows an additional numerical comparison to further demonstrate the advantages of the proposed EHK method compared with other methods. Three metrics were utilized in the comparison: (1) RRMSE, (2) RMAE, and (3) FEs. The RRMSE and RMAE served as accuracy indicators, measuring the overall and local errors of the surrogate models in relation to the validation data, respectively. The FE metric represented the computational cost associated with hyperparameter optimization, with higher values indicating a more resource-intensive process. Since the EHK, ECK, MHK, and Kriging models are Bayesian-based methods, their unknown hyperparameters were optimized

by maximizing the likelihood function using a genetic algorithm solver. Conversely, the LRMFS model, based on polynomial regression, estimated its parameters directly through the minimization of the sum of squared errors [37], thereby bypassing a tuning process. Consequently, the FE metric was not applicable to the LRMFS model. Additionally, the estimated scaling factors $\rho$ of different resulting models were also presented.

**Table 5.** The numerical comparison of different surrogate models.

|  | EHK | ECK | LRMFS | MHK1 | MHK2 | Kriging |
|---|---|---|---|---|---|---|
| RRMSE | 0.0005 | 0.1408 | 2.5575 | 0.14260 | 0.22910 | 0.4121 |
| RMAE | 0.0011 | 0.2670 | 5.6702 | 0.82170 | 0.90930 | 1.8390 |
| FEs | 12,510 | 28,605 | --- | 12,966 | 12,920 | 5420 |
| $\rho$ | $[-4.993, 4.995]$ | $[-4.892, 4.994]$ | $[0.270, 0.387]$ | $[0.833, 0.725]$ | $[1.200, 0.816]$ | --- |

The findings demonstrate that the EHK method excelled in terms of accuracy while maintaining lower FE values compared to other Bayesian-based modeling methods. Although the LRMFS model exhibited the fastest computational speed, it displayed the poorest performance among the models investigated.

In conclusion, the one-dimensional example, with its limited number of HF samples, provided initial evidence of the superiority of the proposed EHK method over other techniques such as ECK, LRMFS, Kriging, and MHK. The EHK method showcased its ability to effectively incorporate multiple non-level LF models, enabling the generation of accurate HF models while keeping the computational cost of model processing at a reasonable level. However, it is worth noting that further investigations are necessary to explore the impact of incorporating multiple LF models, dimensionality, and the size of HF datasets in greater depth. These aspects will be thoroughly examined in subsequent comprehensive studies.

### 4.2. Multi-Dimensional Examples

#### 4.2.1. Effect of Incorporating Multiple LF Datasets

In this section, an extensive evaluation of the proposed EHK method was conducted through the Currin example. The evaluation primarily focused on assessing the computational cost, specifically in terms of FEs, and its relationship with the number of LF datasets. To ensure the results' generality, the modified two-dimensional Currin function was utilized for this analysis. This choice enabled generating arbitrary sampling points and their corresponding LF responses. Furthermore, the modified Currin function featuring two variables facilitated a clear and convenient observation of model behaviors, aligning with the study's objectives. The analysis addressed several critical aspects simultaneously, including high dimensionality, the integration of a substantial number of LF datasets, model accuracy, and computational expense. The HF and LF models are given in Equations (27) and (28) [37,48], respectively. Multiple coefficient values A, B, and C were randomly sampled in the range of 0 to 1, resulting in the creation of eight distinct LF models. These LF models were designed to capture the overall trends exhibited by the HF model, and details can be found in Table A1.

$$f_{HF}(\boldsymbol{x}) = \left[1 - \exp\left(-\frac{1}{2x_2}\right)\right] \frac{2300x_1^3 + 1900x_1^2 + 2092x_1 + 60}{100x_1^3 + 500x_1^2 + 3x_1 + 20} \tag{27}$$

$$\begin{aligned} f_{LF}(\boldsymbol{x}) = \quad & A[(x_1 + 0.05x_2 + 0.05)f_{HF}(\boldsymbol{x}) + (x_1 + 0.05 \times \max(0, x_2 - 0.05))f_{HF}(\boldsymbol{x})] \\ & + B[(x_1 - 0.05x_2 + 0.05)f_{HF}(\boldsymbol{x}) + (x_1 - 0.05 \times \max(0, x_2 - 0.05)] \\ & + C(-5x_1 - 7x_2^2) \end{aligned} \tag{28}$$

$$\boldsymbol{x} = (x_1, x_2)^{\mathbf{T}}, \ \boldsymbol{x} \in [0, 1]^2,$$
$$A, B, C \in [0, 1]$$

Figure 4 demonstrates the responses of $f_{HF}(x)$ and $f_{LF}(x)$ when A = 0.12, B = 0.44, and C = 0.85. The proposed EHK method underwent evaluation in seven distinct testing scenarios, each involving the construction of HF models. These HF models were created using a consistent set of HF samples but varied in the number of incorporated LF datasets. Detailed descriptions and numerical outcomes of these investigations are provided in Table 6. In these investigations, three metrics were considered: (1) $n_{hyp}$, (2) FEs, and (3) RRMSE.

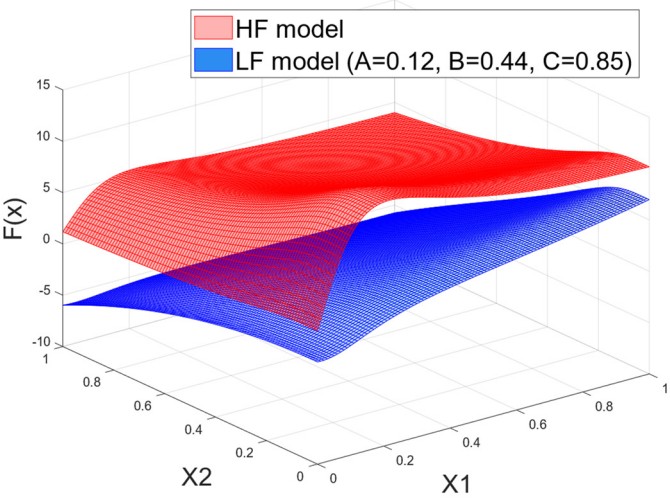

**Figure 4.** HF and LF models of the Currin function.

**Table 6.** Description of testing cases and numerical results.

| Case | Numb. LF Models | EHK | | | | ECK | | | | LRMFS | | |
|---|---|---|---|---|---|---|---|---|---|---|---|---|
| | | $n_{hyp}$ ($n_c$) | FEs | RRMSE | Time (s) | $n_{hyp}$ ($n_c$+$n_s$) | FEs | RRMSE | Time (s) | $n_{hyp}$ | RRMSE | Time (s) |
| 1 | 2 | 2 | 13,510 | 0.3931 | 8.1 | 2 + 2 | 13,610 | 0.4700 | 31.3 | - | 1.2291 | $4 \times 10^{-3}$ |
| 2 | 3 | 2 | 15,220 | 0.2952 | 5.1 | 2 + 3 | 17,410 | 0.5201 | 37.4 | - | 1.2287 | $8 \times 10^{-3}$ |
| 3 | 4 | 2 | 14,080 | 0.2946 | 13.1 | 2 + 4 | 114,210 | 0.4374 | 59.8 | - | 1.2285 | $5 \times 10^{-3}$ |
| 4 | 5 | 2 | 14,270 | 0.2953 | 18.4 | 2 + 5 | 133,210 | 0.3954 | 120.5 | - | 1.2288 | $2 \times 10^{-3}$ |
| 5 | 6 | 2 | 13,510 | 0.2941 | 11.7 | 2 + 6 | 152,210 | 0.3900 | 68.44 | - | 1.2290 | $9 \times 10^{-3}$ |
| 6 | 7 | 2 | 12,750 | 0.2954 | 10.6 | 2 + 7 | 171,210 | 0.3551 | 69.2 | - | 1.2283 | $8 \times 10^{-4}$ |
| 7 | 8 | 2 | 12,370 | 0.2943 | 11.3 | 2 + 8 | 190,210 | 0.9072 | 90.6 | - | 1.2281 | $9 \times 10^{-4}$ |

The impact of the number of LF datasets on the constructed models was examined by incrementally increasing the number of LF datasets from 2 to 8. The HF and LF samples were generated using the LHS method with $n_{HF} = 19$ points and $n_{LF} = 200$ points. An additional 10,000 HF samples were utilized for validation purposes to assess the accuracy of the resulting approximation models. Then, the EHK model was compared with the ECK and LRMFS models. It is worth noting that the FE metric was not considered for the LRMFS model, and the evaluation of the LRMFS model solely relied on the RRMSE metric.

The findings, as presented in Table 6 and Figure 5, unequivocally establish the superior performance of the EHK method. Of particular note is the contrast between the EHK model and the ECK model, wherein the EHK model exhibited a consistent number of hyperparameters at 2, equivalent to the dimensions of the dataset, regardless of the expanding number of LF datasets. This observation underscored the efficiency and adaptability of the EHK method in handling complex LF datasets. Furthermore, the EHK model consistently maintained a stable number of FEs and computational time, approximately 13,672 s and 11.32 s, respectively, irrespective of the number of LF models. In sharp contrast, the ECK model experienced a substantial surge in FEs and computational time as the number of LF models increased. To illustrate, the EHK model realized an impressive cost reduction of

around 80% with two LF models and a substantial 94% cost reduction with eight LF models in comparison to the ECK model. This striking outcome highlighted the EHK model's capacity to significantly reduce the computational cost of model tuning while concurrently preserving high levels of accuracy. It is important to note that the LRMFS model also saw an increase in hyperparameters due to LF models. However, these hyperparameters were directly estimated, obviating the necessity for an optimization process. Consequently, they were excluded from the computational cost comparison. While LRMFS emerged as the most cost-effective method, it lagged in accuracy compared to other models across the tested scenarios.

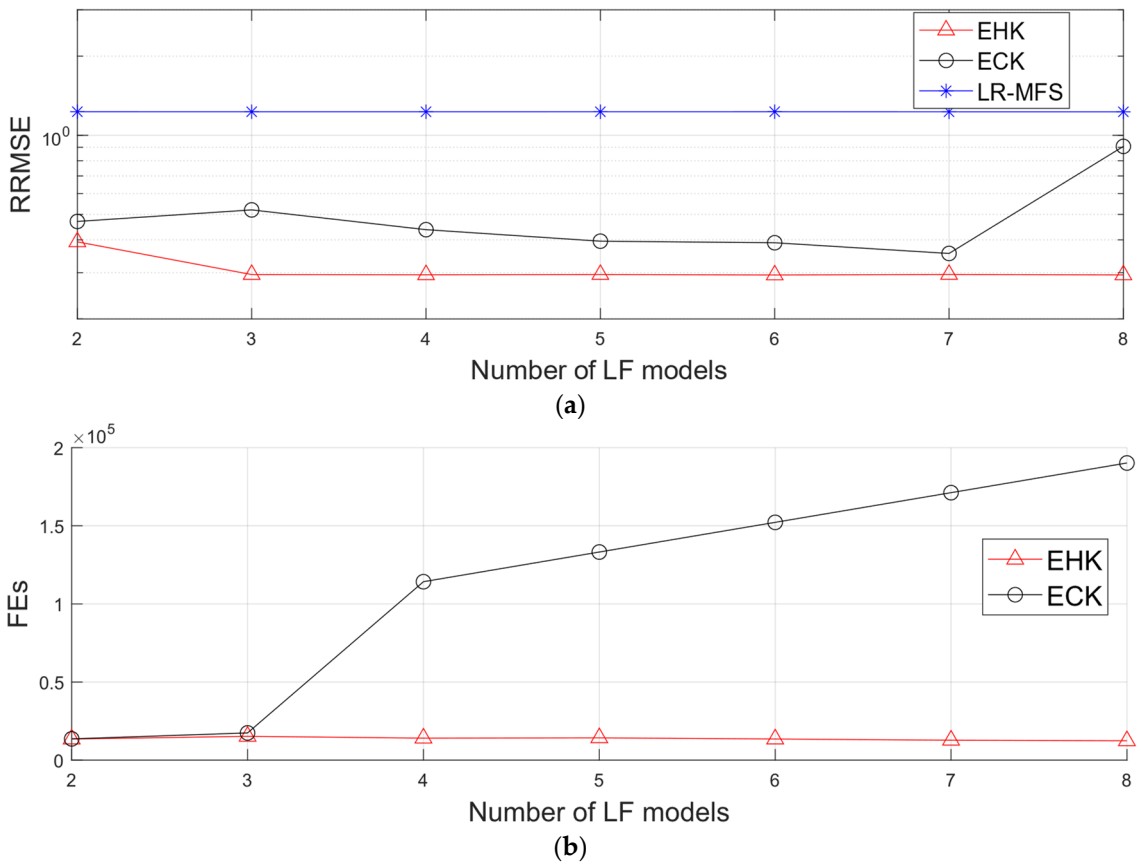

**Figure 5.** Performance comparison of different models regarding the number of LF models incorporated: (**a**) global error; (**b**) cost of hyperparameter optimization.

Figure 5a further illustrates the general improvement in accuracy with an increasing number of LF models, with the exception of the LRMFS model. Among all the cases investigated, the EHK method consistently achieved the highest accuracy, while the LRMFS method consistently produced the lowest. Figure 5b graphically represents the computational costs of the EHK and ECK models in relation to the number of LF models, reinforcing the quantitative results detailed in Table 6. These findings firmly establish the efficiency and scalability of the EHK method, solidifying its position as a highly favorable choice for multi-fidelity modeling. These empirical results robustly corroborate the theoretical framework presented in Equation (20), highlighting the advantages of the EHK method in terms of reducing computational costs while concurrently enhancing accuracy, thus distinguishing it from the existing methods.

### 4.2.2. Effect of HF Data Size and Dimensionality

The versatility of the EHK method was rigorously examined across eight distinct testing scenarios, each varying in dimensionality and the number of LF models incorporated.

The primary objective of this investigation was to evaluate the influence of the number of HF sample points and dimensionality on the model's accuracy, while also conducting a comparative assessment between the EHK models, the ECK models, and the LRMFS models. Comprehensive information regarding the eight numerical examples is available in Table 7, with mathematical expressions for these examples listed in Table A2.

**Table 7.** Features of eight test cases [38].

| Test | No. of Dimension | No. of LF Models | Initial Sample Sets |
|------|------------------|------------------|---------------------|
| 1 | 1 | 2 | 3HF\|40LF ($20 \times 2$) |
| 2 | 1 | 3 | 3HF\|60LF ($20 \times 3$) |
| 3 | 1 | 2 | 3HF\|40LF ($20 \times 2$) |
| 4 | 1 | 2 | 3HF\|40LF ($20 \times 2$) |
| 5 | 1 | 2 | 3HF\|40LF ($20 \times 2$) |
| 6 | 2 | 2 | 9HF\|100LF ($50 \times 2$) |
| 7 | 2 | 2 | 9HF\|100LF ($50 \times 2$) |
| 8 | 4 | 2 | 21HF\|400LF ($200 \times 2$) |

For each testing case, the training sets of the HF and LF samples were randomly generated using the LHS method. The number of validation points varied across the examples, with 100 points for the one-dimensional case and 10,000 points for the remaining numerical cases. To ensure the accuracy of the resulting models, the LHS method was repeated 50 times for each number of HF samples, thereby mitigating the influence of the experimental design on the results. The average relative root mean square error (avgRRMSE) across all these random runs was used to evaluate the overall error of the constructed models in relation to the size of the HF dataset. Figure 6 presents a visual representation of the average RRMSE for the different models as the number of HF samples varies.

Figure 6 reveals the consistent superiority of the EHK models in improving the average RRMSE as the number of HF samples increases. Specifically, the proposed EHK methods exhibited significant advantages over other models in tests 1, 2, 3, 4, 5, and 8. In tests 6 and 7, it was more challenging to definitively determine the winner between the EHK and LRMFS models. Nonetheless, even in these cases, the EHK models consistently showed smaller averaged RRMSE values compared to the LRMFS models, especially when dealing with a limited number of HF samples. These compelling findings, derived from multiple testing scenarios, validated the efficacy of the proposed EHK model in addressing data fusion challenges involving high-dimensional data and multiple LF datasets. The EHK model exhibited remarkable potential for enhancing accuracy and performance in such intricate scenarios.

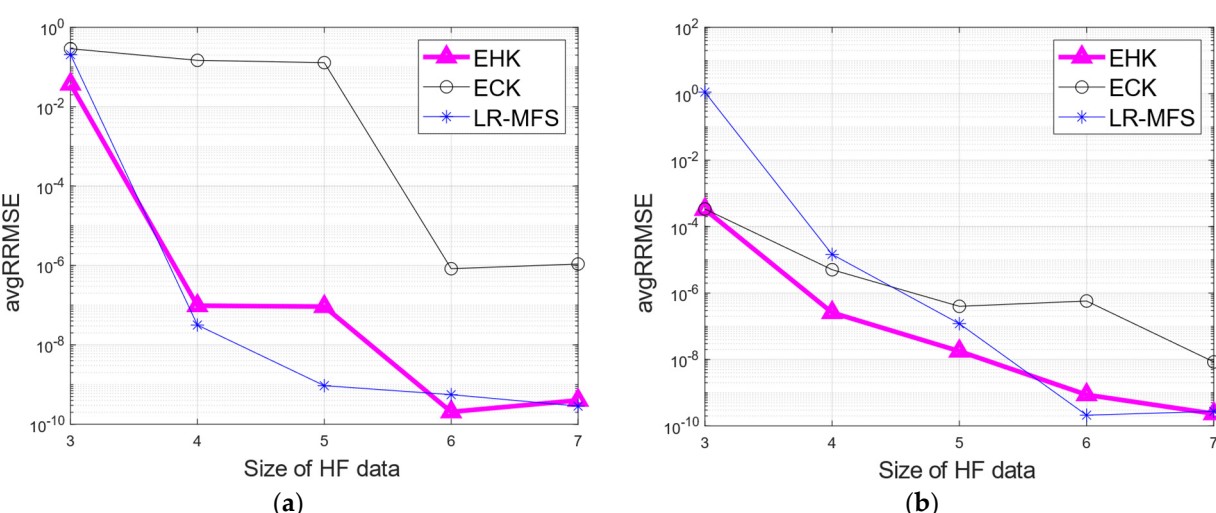

**Figure 6.** *Cont.*

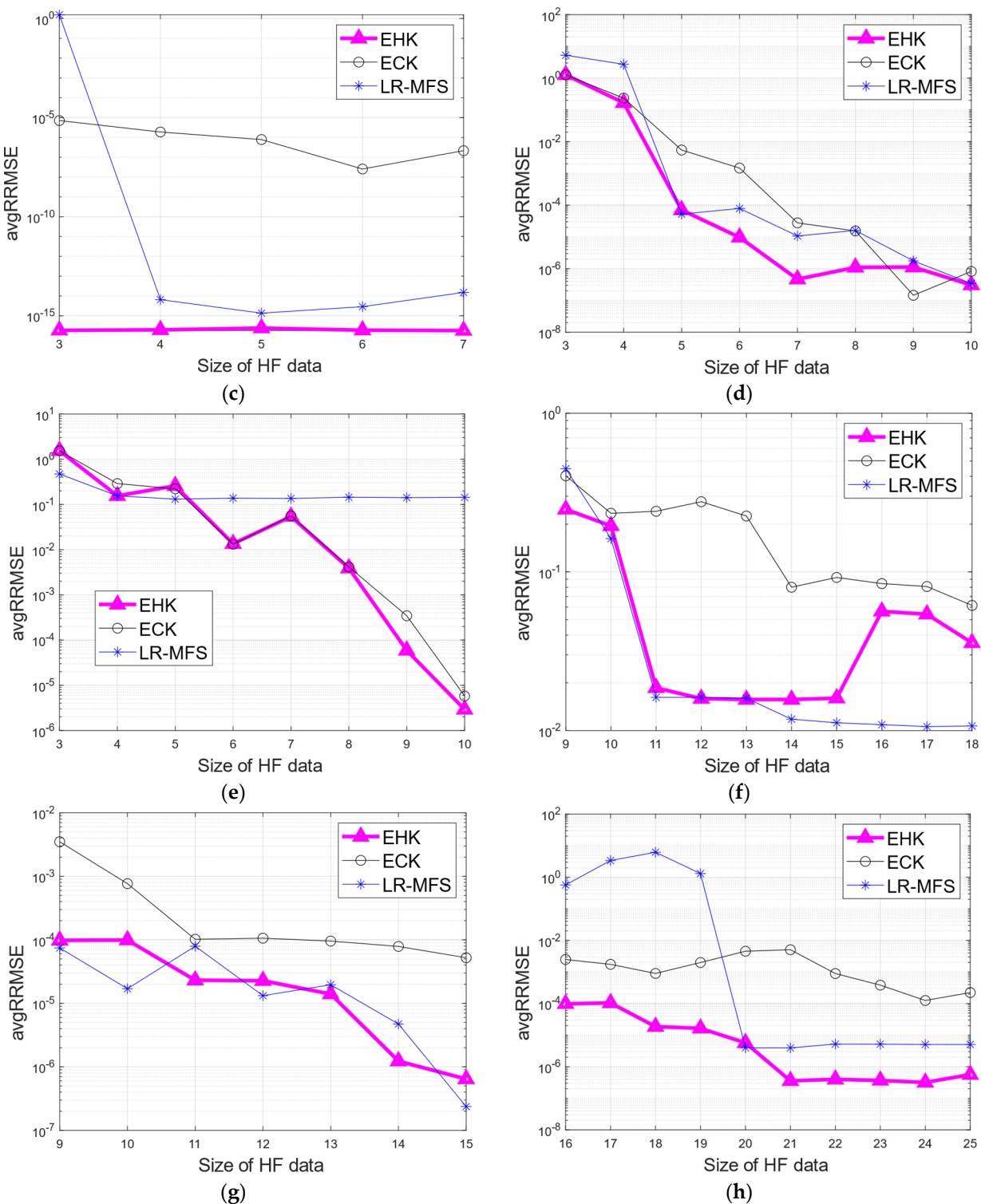

**Figure 6.** Error comparison of surrogate models regarding the number of HF samples. (**a**) Test 1, (**b**) test 2, (**c**) test 3, (**d**) test 4, (**e**) test 5, (**f**) test 6, (**g**) test 7, and (**h**) test 8.

*4.3. Engineering Examples*

4.3.1. Approximation of the Fluidized Bed Process

Dewettinck et al. [49] conducted a comprehensive study involving a physical experiment and the development of associated computer models aimed at predicting the steady-state thermodynamic operation of a GlattGPC-1 fluidized bed unit. This unit com-

prised a base featuring a screen and air jump, complemented by coating sprayers positioned along the unit's sides. Quian and Wu [23] introduced a Bayesian hierarchical Gaussian process model designed to concurrently analyze both experimental data and computer simulations. Chen et al. [39] contributed to the analysis of the same dataset, offering non-hierarchical multi-model fusion techniques, including WS, PC-DIT, and PC-CSC. In a parallel effort, Zhang et al. [37] applied their proposed LRMFS approach to address the same problem.

In this section, a comprehensive comparison was conducted to evaluate the effectiveness of the proposed EHK method against six existing MFSM methods, namely ECK, LRMFS, WS, PC-DIT, and PC-CSC, utilizing the reported datasets from the fluidized bed process example. Furthermore, the influence of incorporating different LF datasets was explored on the outcomes of the analysis.

The focus of this study was the determination of temperature ($T_2$) at the steady-state thermodynamic operational point within a fluidized bed process. This temperature is subject to variation due to six key variables: the humidity, the room temperature, the temperature of the air from the pump, the flow rate of the coating solution, the pressure of atomized air, and the fluid velocity of the fluidization air. The foundational data for the investigation has been sourced from prior research [23,49], which meticulously documented these six input variables and the corresponding responses. This dataset was collected through a combination of experimental measurements and computer simulations, encompassing a diverse range of 28 distinct process conditions. Notably, the experiments featured the use of distilled water as the coating solution at room temperature. The analysis placed a particular emphasis on several output variables, which were categorized into four fidelity levels: $T_{2,exp}$, $T_{2,3}$, $T_{2,2}$, and $T_{2,1}$. Here, $T_{2,exp}$ was designated as the highest-fidelity experimental response. In contrast, $T_{2,3}$ reflected the most precise simulation, incorporating adjustments to account for heat losses and inlet airflow. $T_{2,2}$ offered intermediate accuracy, considering adjustments associated with heat losses, while $T_{2,1}$ represented the lowest level of accuracy, devoid of any adjustments related to either heat losses or inlet airflow.

In previous research, Quian and Wu [23] limited their analysis to data derived from $T_{2,2}$ and $T_{2,exp}$. However, Chen et al. [39] took a more comprehensive approach by incorporating additional data from a less accurate model, $T_{2,1}$, to enhance their final predictions compared to the work of Quian and Wu. The primary focus of this study was the prediction of the high-fidelity dataset $T_{2,exp}$ using the EHK method, with datasets $T_{2,1}$ and $T_{2,2}$ serving as the low-fidelity counterparts. All 28 runs of $T_{2,1}$ and $T_{2,2}$, along with the remaining 20 runs of $T_{2,exp}$, were utilized for training the approximation models. To validate the resulting models, eight specific physical experiment runs, corresponding to $T_{2,exp}$ (specifically runs 4, 15, 17, 21, 23, 25, 26, and 28, as previously employed by Quian and Wu) were reserved as shown in Figure 7. The correlation coefficients between the computer simulation datasets, $T_{2,1}$ and $T_{2,2}$, and the experiment dataset $T_{2,exp}$ were reported as 0.9754 and 0.9774, respectively. These values indicated that the dataset $T_{2,2}$ exhibited a slightly stronger correlation with the experimental dataset $T_{2,exp}$ compared to $T_{2,1}$.

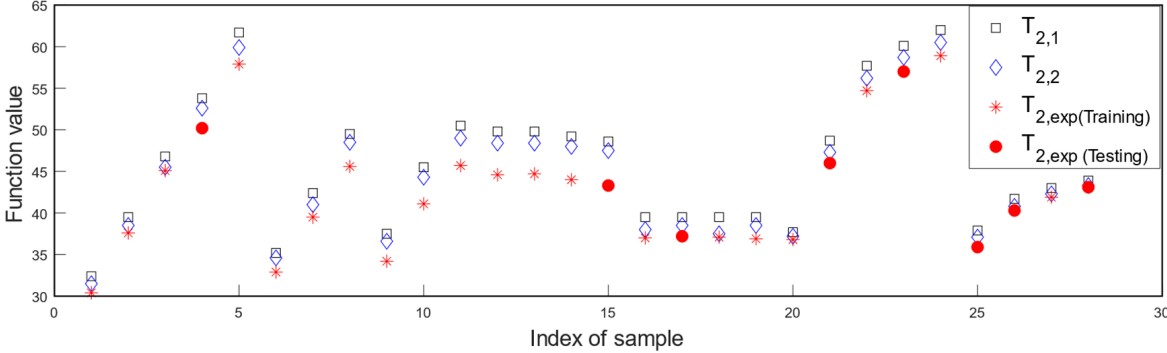

**Figure 7.** Function value of different input datasets for model construction.

Figure 8 demonstrates the predictions of the EHK, ECK, and LRMFS models at eight reserved validation points against the experimentally observed steady-state outlet air temperatures. For the majority of validation points, both the EHK and ECK models closely aligned with the $y = x$ line, indicating a high degree of accuracy in their predictions, which closely matched the observed values. In contrast, the predictions of the LRMFS model exhibited lower accuracy.

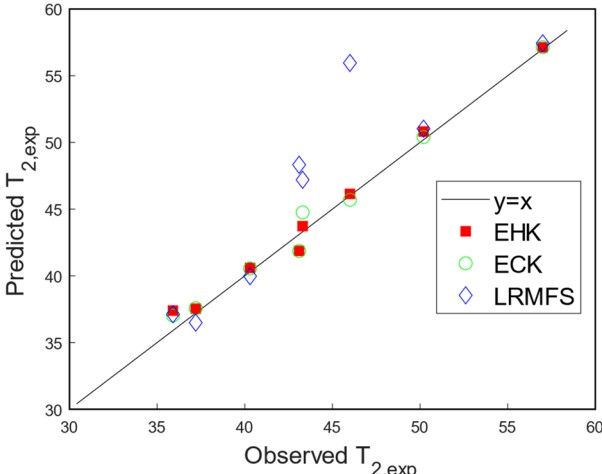

**Figure 8.** Observed versus predicted steady-state outlet air temperature in eight untrained conditions from various methods.

For a more in-depth assessment of the model performance, a comprehensive comparison of MFSM models with respect to various metrics was provided in Table 8. In addition to comparing the proposed EHK model with the ECK and LRMFS models, previous works from Chen et al. were also included for a comprehensive evaluation.

**Table 8.** Metrics of MFSM models in prediction of T2,$_{exp}$ at eight reserved validation points.

| Methods | EHK | ECK | LRMFS | Chen's | | |
|---|---|---|---|---|---|---|
| | | | | WS | PC-DIT | PC-CSC |
| $\rho_1, \rho_2$ | [0.1806, 0.7715] | [0.1813, 0.7718] | [0.278, 5.0078] | / | / | / |
| $n_{hyp}$ | 6 | 20 | / | 21 | 19 | 19 |
| FEs | 25,395 | 97,320 | / | / | / | / |
| | | +73.90% | / | / | / | / |
| RRMSE | 0.0937 | 0.1084 | 0.6116 | 0.1065 | 0.0990 | 0.0996 |
| | / | +13.56% | +84% | +12.01% | +5.35% | +5.92% |
| RMAE | 0.1878 | 0.2976 | 1.4323 | / | / | / |
| | / | +36.89% | +86.88% | / | / | / |

In this example, the EHK model demonstrated a substantial reduction in the number of tuning hyperparameters compared to other models. It consistently maintained a minimal set of six hyperparameters, corresponding to the dimensions of the training data, regardless of the number of incorporated datasets. In contrast, the ECK, WS, PC-DIT, and PC-CSC models had a significantly higher number of hyperparameters, often exceeding that of the EHK model by more than three times. This substantial difference in the number of hyperparameters led to a considerable increase in computational cost, measured in FEs, particularly for the ECK model, which needed to estimate 20 hyperparameters. Consequently, the EHK model achieved a notable 73.9% reduction in computational cost compared to the ECK model, highlighting its computational efficiency. This efficiency can be attributed to the advanced framework and model-processing algorithm inherent to the EHK method.

Furthermore, estimated scaling factors, denoted as $\rho_1$ and $\rho_2$, associated with datasets $T_{2,1}$ and $T_{2,2}$, respectively, in different models were provided. It was observed that the values of $\rho_2$ were considerably larger than $\rho_1$ in all EHK, ECK, and LRMFS models, indicating that dataset $T_{2,2}$ was assigned greater weight due to its higher fidelity. This correlation between the values of scaling factors aligned with the reported values of the correlation coefficients between the simulation datasets $T_{2,1}$ and $T_{2,2}$, and the experimental dataset $T_{2,exp}$. Lastly, the metrics of RRMSE and RMAE indicated that the EHK model attained a slightly higher level of accuracy compared to the other approaches. In contrast, the LRMFS model exhibited the poorest performance, despite having the lowest cost of model processing. The EHK model's reduction in the error metric RRMSE ranged from a maximal value of 84% to a minimal value of 5.35% when compared with the LRMFS and PC-DIT models, respectively. Regarding the RMAE metric, the comparison involved only the EHK, ECK, and LRMFS methods since no estimation of the metric for the WS, PC-DIT, and PC-CSC models was reported in the original work. The EHK model's reductions in RMAE over the ECK and LRMFS models were 36.89% and 86.88%, respectively. These findings underscored the EHK model's enhanced accuracy and efficiency in this particular analysis.

In conclusion, the EHK method has demonstrated its prowess as a formidable solution for approximating the fluidized bed process, even when dealing with six-dimensional and three-fidelity-level datasets. Its superior performance, when compared to existing approaches, underscored its potential to revolutionize the field of engineering. EHK's precision and efficiency position it as a valuable tool for addressing complex engineering design problems, offering innovative solutions, and driving progress within the industry.

4.3.2. Generation of Aerodynamic Models for an eVTOL Vehicle for Urban Air Mobility

In this application, the EHK method was employed to develop aerodynamic models of six aerodynamic coefficients for an eVTOL-KP2 aircraft [50,51], including: drag coefficient ($C_D$), lift coefficient ($C_L$), pitching moment coefficient ($C_m$), side force coefficient ($C_Y$), rolling moment coefficient ($C_l$), and yawing moment coefficient ($C_n$). The tridimensional depiction of the KP-2 design is illustrated in Figure 9, accompanied by a comprehensive overview of its design characteristics provided in Table 9.

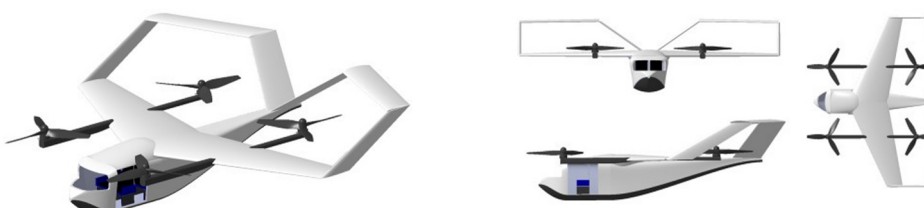

**Figure 9.** eVTOL-KP2 aircraft design.

**Table 9.** Design features of KP-2 aircraft.

| Metrics | Value |
| :---: | :---: |
| Empty weight | 8.2 kg |
| Wingspan | 2 m |
| Wing platform area | 0.705 m$^2$ |
| Aspect ratio | 7.19 |
| Mean aerodynamic chord | 0.2995 |
| Fuselage length | 1.4 m |
| Cruise speed | 25 m/s |

The input data consist of two independent variables, representing simulated flight conditions: the angle of attack and the sideslip angle, denoted as $\alpha$ and $\beta$, respectively. These variables were constrained within the ranges of $-20° \leq \alpha \leq 30°$ and $0° \leq \beta \leq 50°$, while the simulation was conducted at a fixed velocity of 25 m/s. The initial training

datasets were generated using three distinct analysis methods, each offering varying levels of fidelity. The LHS method was employed to create sampling plans for the HF and LF analyses. The HF dataset encompassed 80 sample points produced through CFD analysis using ANSYS-Fluent 2020 R2 software, while the LF datasets were generated using the HETLAS [52], AVL [53], and XFLR5 [54] analysis tools, resulting in three LF datasets, each comprising 1300 sample points. It is worth noting that while HETLAS, AVL, and XFLR5 are considered low-fidelity analysis tools, they offer the advantage of producing a large volume of data points, spanning a wide spectrum of flight conditions at a relatively modest computational cost. The EHK method was applied in this study to construct surrogate models of aerodynamic coefficients using varying numbers of CFD samples, ranging from 9 to 80 data points, as detailed in Table 10. To ensure the accuracy of the constructed models and mitigate the influence of the design of experiments, the sampling was repeated 10 times for each number of CFD samples, randomly generating different CFD sampling plans from the pool of 80 generated CFD data points.

**Table 10.** Domains and analysis methods for aerodynamic coefficients.

| Analysis Tool | Variables | | Initial Sample Points |
|---|---|---|---|
| | $\alpha$ | $\beta$ | |
| HETLAS | $[-20, 30]$ | $[0, 50]$ | 1300 |
| AVL | $[-20, 30]$ | $[0, 50]$ | 1300 |
| XFLR5 | $[-20, 30]$ | $[0, 50]$ | 1300 |
| Fluent | $[-20, 30]$ | $[0, 50]$ | From 9 to 80 |

Figure 10 shows a 1/4-scale model of a KP-2 aircraft for simulation, and an unstructured mesh with 10,970,498 cells was applied for the CFD model. For the CFD analysis, a 3D model of the KP-2 aircraft was discretized into the numerical domain using the unstructured meshing method. The surface mesh was generated through Ansys Fluent meshing, employing an unstructured triangle mesh. The dimensions of the computational domain, depicted in Figure 10a, were set to 20 times the length of the fuselage in all directions. In each simulation case, the aircraft was oriented to the respective $\alpha$ and $\beta$. The inlet boundary condition enforced a constant velocity on the semi-spherical face of the domain and the lateral wall of the cylinder. Pressure outlet boundary conditions were applied at the end of the domain.

To enhance flow accuracy near the wall boundary layer, a 20-layer prism mesh with a Yplus value set to 1 was applied on top of the surface mesh. The Yplus value dictates the mesh's ability to capture the boundary layer flow phenomena, representing the distance of the first layer of the boundary layer mesh. This structured mesh was then extruded into a volume mesh using a tetrahedral mesh configuration. Subsequently, for increased convergence rates and reduced mesh size, the unstructured mesh was converted into a polyhedral mesh using ANSYS Fluent 2020 R2 meshing, as illustrated in Figure 10b.

The three-dimensional compressible fluid flow is simulated using the Reynolds-averaged Navier–Stokes (RANS) equations, assuming incompressible flow due to the low Mach number achieved by the aircraft. For stall conditions, where the high angle of attack may lead to boundary layer separation on certain zones of the aircraft surface, an appropriate turbulence model capable of accounting for such separation is necessary. In this analysis, the shear stress transport k–$\omega$ turbulent model was employed to simulate the aerodynamics of the wing and the entire aircraft. The set of governing equations was solved in Ansys Fluent using the SIMPLE algorithm for pressure–velocity decoupling [55]. The simulations were carried out on a computer featuring a configuration of an Intel® Xeon® W-2265 CPU @3.50 GHz, 12 cores, and 128 GB of RAM. The total computational time for the AVL and HETLAS cases amounted to a couple of hours, demonstrating efficient processing. In contrast, the CFD cases necessitated approximately 320 h for completion, indicating a significantly longer duration due to their computational complexity.

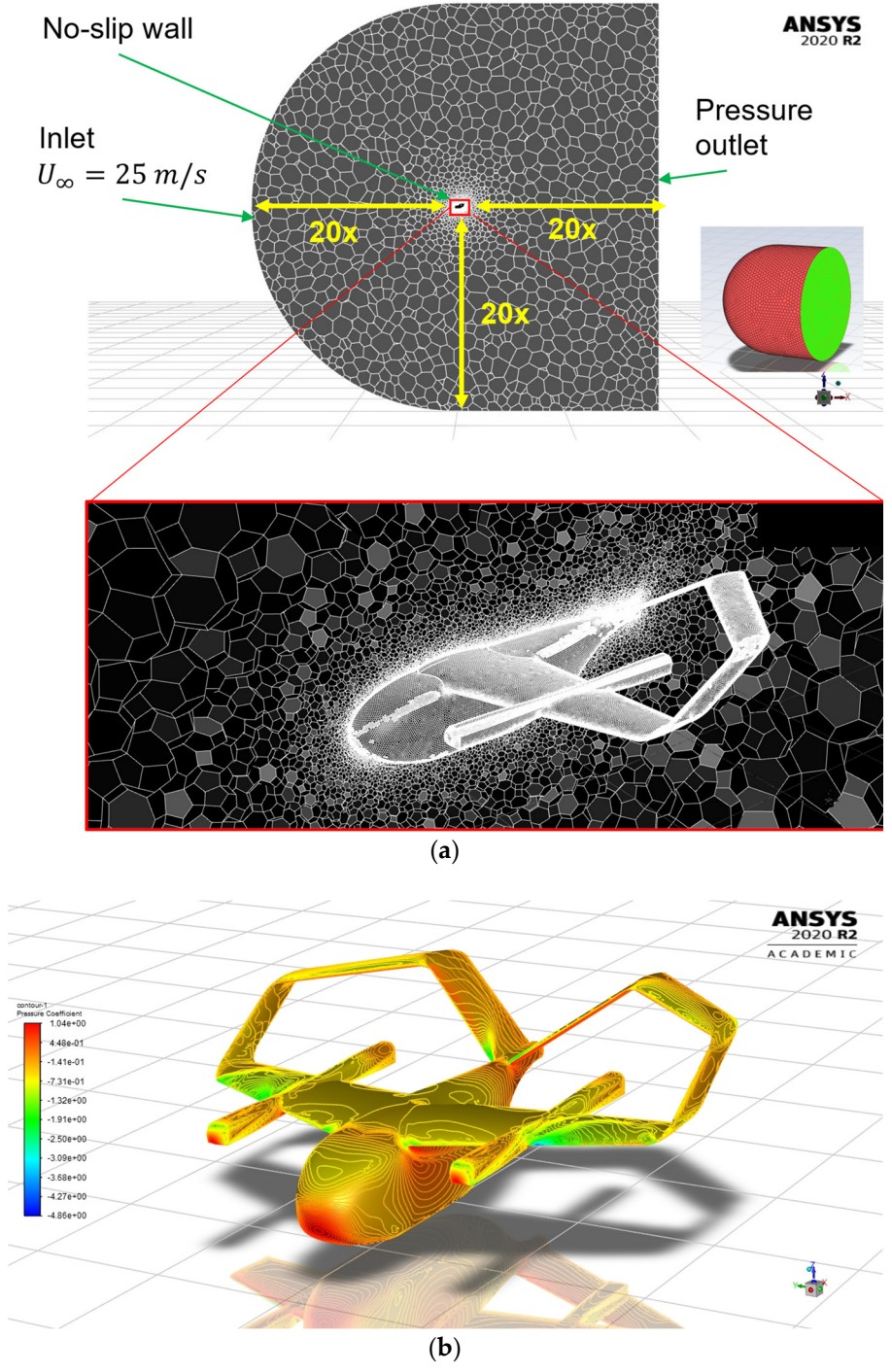

**Figure 10.** (**a**) Presentation of boundary conditions, far-field and prism boundary layer mesh of fuselage and wing. (**b**) Visualization of simulation results depicting pressure distribution on the surface.

Figure 11 illustrates the initial sampling plan with 9 CFD points. Moreover, an additional 30 CFD points were randomly selected to validate the accuracy of the resulting models. An error comparison in terms of average RRMS between the EHK, ECK, and LRMFS models for aerodynamic coefficients regarding the number of CFD data points is shown in Figure 12.

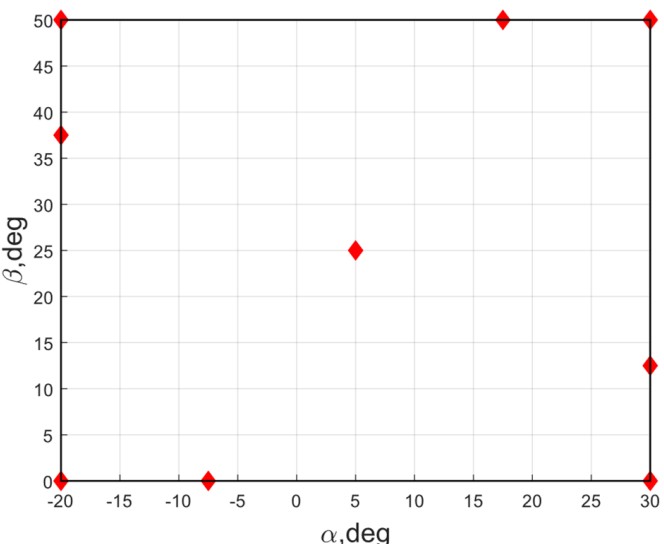

**Figure 11.** Initial sampling plan with nine CFD points in red diamonds.

Generally, it is observed that the accuracy of the aerodynamic models by different methods was improved when the number of CFD points increased. The EHK models exhibited the best performances in approximating aerodynamic coefficients compared to the other models. The EHK and LRFMS models performed better than the ECK model when the number of CFD samples was less than 24 points. However, the LRMFS model's accuracy was "saturated" after 24 CFD sample points. In contrast, the accuracy of the EHK and ECK models continued to be significantly improved when the number of CFD samples was higher than 24 points. With more than 24 CFD samples, the EHK model was slightly better than the ECK models in the approximations of $C_D$, $C_L$, $C_m$, $C_Y$. The EHK models significantly outperformed the ECK models for the approximations of $C_n$ and $C_l$. Furthermore, the computational times recorded for the EHK, ECK, and LRMFS models across all testing cases were as follows: 15.388 s, 68.830 s, and 3.081 s, respectively.

Based on the previous investigations, it was consistently observed that the EHK model exhibited a lower computational cost during the model processing compared to the ECK model, primarily attributed to its reduced number of tuning hyperparameters. In the same training conditions, the EHK models achieved a remarkable reduction in computational time by 73% while maintaining higher levels of accuracy compared to the ECK models. The importance of this efficiency gain becomes even more apparent when considering the scalability and versatility of the EHK method. Whether applied on typical computer setups or extended to tackle more complex engineering tasks featuring voluminous datasets, high dimensionality, and the need for iterative model constructions, the EHK method's efficiency and computational savings are undeniable.

Figure 13 provides cross-sectional views of the resulting response surfaces for the lift, drag, and pitching moment coefficients at a constant sideslip angle of $\beta = 0$. These surfaces were constructed using 24 CFD samples and 2600 LF samples. The accuracy comparison between the EHK model and other models was performed over ten repetitions, each involving 24 CFD samples. Evaluation metrics such as avgRRMSE and avgRMAE are presented in Table 11. Remarkably, the LRMFS models achieved the lowest accuracy, despite their substantially lower computational costs in comparison to the other models. Furthermore, the EHK models demonstrated a slight edge in accuracy over the ECK model while maintaining a lower computational cost with an equal number of CFD training points.

In summary, the proposed EHK method stood out as an exceptionally efficient multi-level MFSM approach, outperforming other state-of-the-art methods, including the LRMFS and ECK models.

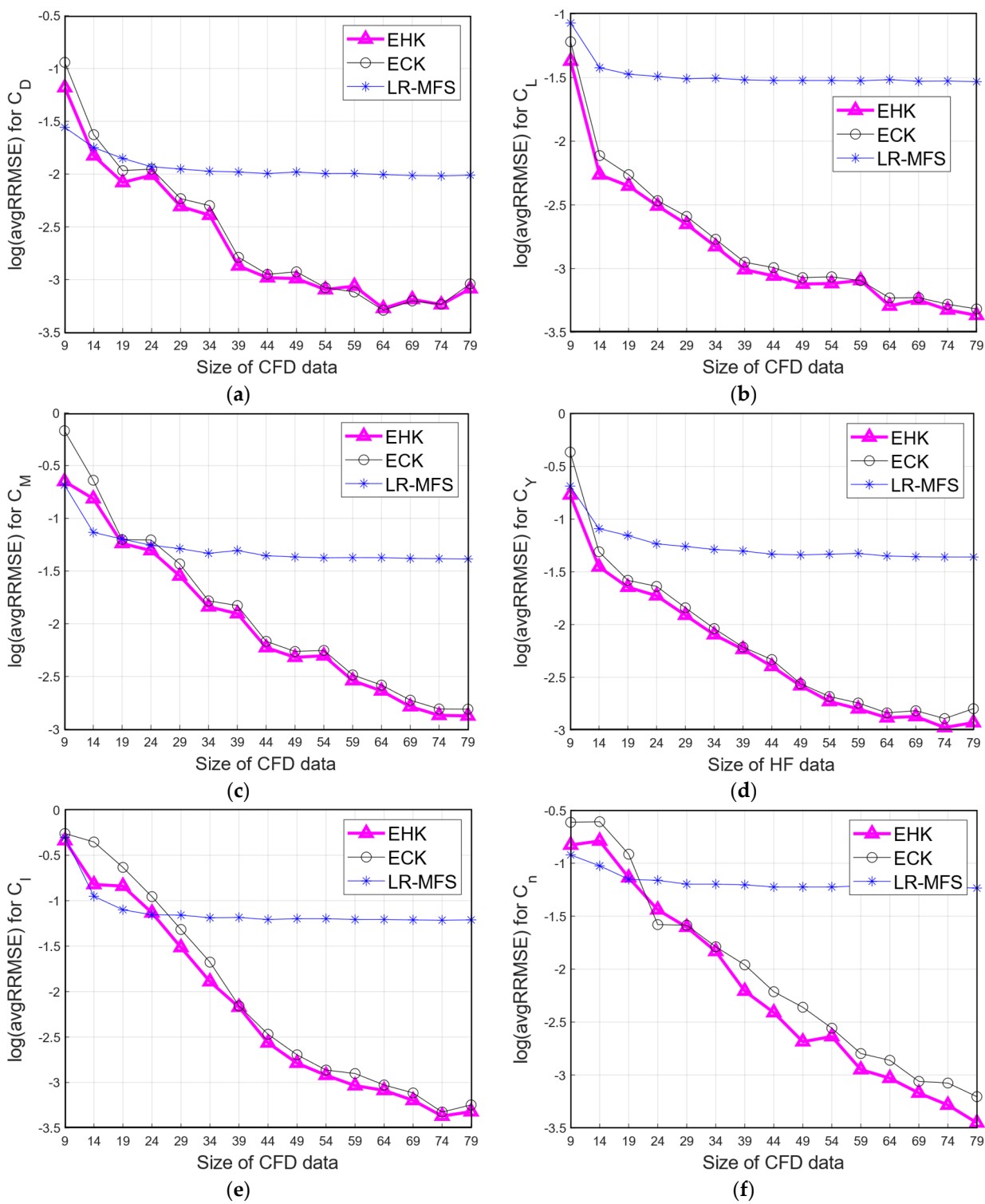

**Figure 12.** Average RRMSE of aerodynamic models from 10 repetitions using different modeling methods regarding the number of CFD samples. (**a**) Drag coefficient, (**b**) lift coefficient, (**c**) pitching moment coefficient, (**d**) side force coefficient, (**e**) rolling moment coefficient, and (**f**) yawing moment coefficient.

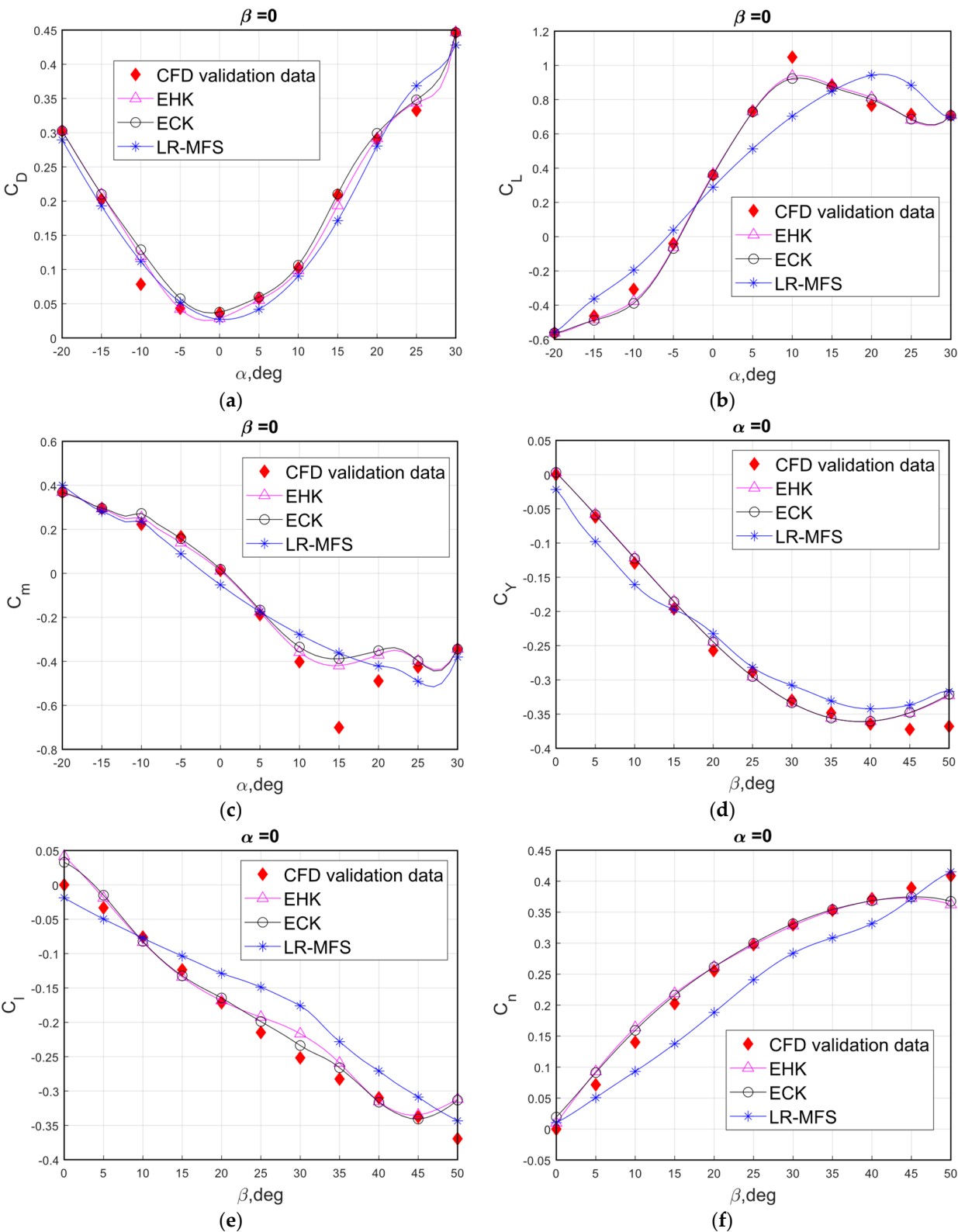

**Figure 13.** Comparison of prediction models (24 CFD samples and 2600 LF samples) with validation data. (**a**) Lift coefficient, (**b**) drag coefficient, (**c**) pitching moment coefficient, (**d**) side force coefficient, (**e**) rolling moment coefficient, and (**f**) yawing moment coefficient.

**Table 11.** Accuracy metrics of approximation models from 10 repetitions regarding 24 CFD samples.

| Coeff | avgRRMSE | | | avgRMAE | | |
|---|---|---|---|---|---|---|
| | **EHK** | **ECK** | **LRMFS** | **EHK** | **ECK** | **LRMFS** |
| $C_D$ | 0.099 | 0.102 | 0.142 | 0.293 | 0.364 | 0.269 |
| | | +3% | +43% | | +24% | −8.2% |
| $C_L$ | 0.0705 | 0.0713 | 0.221 | 0.189 | 0.212 | 0.588 |
| | | +1.1% | +213% | | +12% | +211% |
| $C_m$ | 0.213 | 0.227 | 0.276 | 0.772 | 0.857 | 0.928 |
| | | +6.5% | +29.5% | | +11% | +16% |
| $C_Y$ | 0.148 | 0.151 | 0.284 | 0.340 | 0.354 | 0.384 |
| | | +2% | +92% | | +4% | +13% |
| $C_l$ | 0.220 | 0.255 | 0.314 | 0.456 | 0.445 | 0.600 |
| | | +16% | +43% | | −3% | +32% |
| $C_n$ | 0.195 | 0.201 | 0.302 | 0.339 | 0.301 | 0.492 |
| | | +3% | +55% | | −11% | +45% |

## 5. Discussion

The aim of this research was to address data fusion problems in the field of aerospace engineering where the size of the HF dataset is not excessively large, and the cost of the model processing is manageable in comparison to expensive simulations. In such cases, MFSM methods are commonly employed to combine multiple large-size LF datasets with a smaller HF dataset due to the significant expense associated with obtaining HF data. The primary objective of most MFSM methods is to minimize the number of costly HF samples required to achieve a surrogate model with the desired level of accuracy. While the performance of the proposed EHK model has been proven to be efficient across diverse data scenarios, including multi-fidelity, multi-dimensional, and varying data sizes, the incorporation of large-size LF datasets during the model construction introduces a certain level of computational complexity. This complexity stems from the need to invert the correlation matrix. Furthermore, challenges related to dimensionality, large-size datasets, and a high number of LF models make the estimation of hyperparameters a non-trivial task in high-dimensional optimization.

This article provided evidence that the EHK model significantly reduced the computational cost of the model processing while maintaining the desired level of accuracy. This enables the incorporation of a much larger number of LF datasets with HF data compared to state-of-the-art methods such as EHK, ECK, and LRMFS. Multiple investigations indicated that the EHK model could effectively handle a considerable number of LF models and up to 15,000 HF data points within a couple of hours using the GA optimizer on a computer configuration consisting of an Intel® Xeon® W-2265 CPU @3.50 GHz, 12 cores, and 128 GB of RAM. The computational time required for the model processing can be tolerated in comparison to the time required for running simulations and generating HF data.

In situations where the HF dataset is large in size, the utilization of multi-fidelity modeling methods to construct an accurate surrogate model of the HF data becomes unnecessary. Instead, single-fidelity modeling methods such as RSM, RBF, and ANNs can be directly applied to the HF samples to create a continuous approximation model. Notably, ANN-based modeling methods exhibit high computational efficiency when dealing with large-size data. However, they require a significant amount of HF data to achieve the desired level of accuracy compared to conventional modeling methods [29]. Consequently, selecting the most suitable modeling method to address any given data scenario remains a challenging task.

To overcome the issue of computational efficiency when incorporating multiple large-size datasets, the proposed EHK model incorporates the following critical features:

- The construction of LF models can be executed independently of the construction of discrepancy functions, employing various modeling methods as needed. In this approach, LF datasets can serve as pre-trained models before the training process with

HF data commences. This approach results in a substantial reduction in the size of the correlation matrix and the overall complexity of the model processing. In cases involving large-size LF datasets, ANN-based modeling methods can be utilized to construct LF models, thereby leveraging computational power to handle the challenges associated with such data sizes.

- Scaling factors are transformed into functions of correlation parameters, leading to a reduction in tuning hyperparameters. In the EHK model, hyperparameters are exclusively related to the correlation parameters of the correlation basis function, regardless of the number of LF models integrated. This streamlined approach significantly reduces computational costs when multiple LF models are incorporated, all without increasing the algorithm's complexity.

In conclusion, these pivotal features enhance EHK's adaptability for future advanced modeling methods and bolsters its scalability for tackling larger engineering problems. The proposed EHK modeling method has been demonstrated to be highly efficient for addressing data fusion problems in aerospace engineering, particularly when the size of the HF dataset is not excessively large.

## 6. Conclusions

In this study, a novel EHK surrogate modeling methodology was proposed to construct a global HF model by integrating original HF data and fusing multiple non-level LF datasets. The advantages of the proposed EHK method were validated against several state-of-the-art methods, through comprehensive investigations. These investigations encompassed ten analytical examples and two engineering examples involving the approximation of the fluidized bed process and the generation of aerodynamic models for the KP-2 aircraft. The impact of multiple LF datasets, dimensionality, and the size of the HF dataset were thoroughly examined. The results consistently demonstrated that the proposed EHK method outperformed other conventional approaches in several key aspects:

- Accuracy: The EHK method yielded more accurate approximation models for various testing functions characterized by multi-fidelity, multi-dimensionality, and diverse landscapes, outperforming the state-of-the-art methods.
- Computational cost: The EHK method effectively addressed the challenge of incorporating multiple non-level LF models, achieving a more affordable computational cost while maintaining the desired level of accuracy.
- Aerodynamic modeling: In the specific case of constructing aerodynamic models for the KP-2 aircraft using multi-fidelity aerodynamic data, the EHK models provided superior accuracy compared to the ECK and LRMFS models. Additionally, the EHK model significantly reduced the computational time of the model processing by 73% compared to the ECK model.

In conclusion, the proposed EHK method holds significant promise for applications in aerodynamic problems and surrogate-based design optimization using high-dimensional and multi-fidelity data. It is also well-suited for other research areas where multi-fidelity computational codes are utilized. Building upon the current EHK method, future work will focus on developing a multi-output multi-fidelity modeling approach aimed at enhancing high-fidelity output by leveraging important information from associated low-fidelity outputs. This approach is particularly relevant for constructing surrogate models for physical systems with numerous inputs and outputs, commonly encountered in multidisciplinary design optimization problems.

**Author Contributions:** Conceptualization, V.P. and J.-W.L.; methodology and software, V.P.; validation and investigation, M.T. and T.A.N.; resources, T.A.N.; writing—original draft preparation, V.P.; writing—review and editing, M.T. and T.A.N.; supervision, J.-W.L. All authors have read and agreed to the published version of the manuscript.

**Funding:** This research was supported by the Basic Science Research Program through the National Research Foundation of Korea (NRF) funded by the Ministry of Education (No. 2020R1A6A1A03046811)

and the Korea Agency for Infrastructure Technology Advancement (KAIA) grant funded by the Ministry of Land, Infrastructure and Transport (Grant RS-2022-00143965).

**Data Availability Statement:** The data presented in this study are available on request from the author.

**Acknowledgments:** The authors would like to thank the editors and the reviewers for their comments on an earlier draft of this article.

**Conflicts of Interest:** The authors declare no conflict of interest.

## Glossary

Nomenclature

| | |
|---|---|
| $\alpha$ | Angle of attach, deg |
| $\beta$ | Sideslip angle, deg |
| $\sigma_{\hat{d}}^2$ | Process variance of discrepancy model |
| $\boldsymbol{\rho}$ | Vector of scaling factors |
| $\phi(.)$ | Correlation function, basic functions |
| $\mu$ | Correction factor |
| $\boldsymbol{\theta}$ | Hyperparameter vector of correlation function |
| $C_L$ | Lift coefficient |
| $C_D$ | Drag coefficient |
| $C_Y$ | Side force coefficient |
| $C_m$ | Pitching moment coefficient |
| $C_l$ | Rolling moment coefficient |
| $C_n$ | Yawing moment coefficient |
| A, B, C | Coefficients of Branin functions |
| $f(x)$ | Actual function |
| $\hat{f}(x)$ | Approximation function |
| $\hat{F}_{LF}$ | Matrix of responses of LF models at HF training points |
| L | Number of low-fidelity datasets |
| $m$ | Number of dimensions |
| $N$ | Total number of training samples |
| $n_{LF}, n_{HF}$ | Number of LF and HF training samples |
| $n_s$ | Number of scaling factors |
| $n_c$ | Number of correlation parameter |
| $n_{test}$ | Number of testing points |
| $n_{hyp}$ | Total number of tuning hyperparameters |
| $O$ | Computational complexity |
| $\boldsymbol{r}(x)$ | Correlation vector |
| $\boldsymbol{R}$ | Correlation matrix |
| $\boldsymbol{S}$ | Dataset |
| $\hat{s}^2$ | Predicted error |
| $T_2$ | Temperature at the steady-state thermodynamic operational point with Different fidelity levels, °C. |
| $U_\infty$ | Velocity of free stream, m/s |
| $x, x'$ | Independent variables |
| $\boldsymbol{x}$ | Sampling plan |
| $\boldsymbol{y}$ | Vector of response |
| $\overline{y}$ | Mean of responses |
| $Z_d(.)$ | Gaussian random process of discrepancy |
| Abbreviation | |
| avgRRMSE | Average relative root mean square error |
| EHK | Extended hierarchical Kriging |
| ECK | Extended co-Kriging |
| FEs | Number of function evaluations of the likelihood function |

| GA | Genetic algorithm |
| HF | High-fidelity values |
| MFSM | Multi-fidelity surrogate modeling |
| MHK | Multi-level hierarchical Kriging |
| LRMFS | Linear regression multi-fidelity surrogate |
| LHS | Latin hypercube sampling |
| LF | Low-fidelity values |
| RRMSE | Relative root mean square error |
| RMAE | Relative maximum absolute error |
| STD | Standard deviation |

## Appendix A

**Table A1.** Values of coefficient for generating LF Currin models.

| Coefficient | $f_{LF,1}$ | $f_{LF,2}$ | $f_{LF,3}$ | $f_{LF,4}$ | $f_{LF,5}$ | $f_{LF,6}$ | $f_{LF,7}$ | $f_{LF,8}$ |
|---|---|---|---|---|---|---|---|---|
| A | 0.52 | 0.07 | 0.40 | 0.33 | 0.55 | 0.79 | 0.16 | 0.12 |
| B | 0.40 | 0.08 | 0.17 | 0.19 | 0.06 | 0.53 | 0.92 | 0.44 |
| C | 0.31 | 0.06 | 0.16 | 0.78 | 0.62 | 0.22 | 0.60 | 0.85 |

**Table A2.** Testing functions [35,38,39] used for the investigation in Section 4.2.2.

| Case | LF Response | HF Response | Input Range |
|---|---|---|---|
| 1 | $f_{LF_1}(x) = \sin x + 0.1(x - \pi)^2$ <br> $f_{LF_2}(x) = 1.2 \sin x + 0.1(x - \pi)^2 - 0.2$ | $f_{HF}(x) = \sin x$ | $0 \le x \le 2\pi$ |
| 2 | $f_{LF_1}(x) = 0.5 f_{HF}(x) + 10(x - 0.5) + 5$ <br> $f_{LF_2}(x) = 0.4 f_{HF}(x) - x - 1$ <br> $f_{LF_3}(x) = 0.3 f_{HF}(x) - 10x + 1$ | $f_{HF}(x) = (6x - 2)^2 \sin(12x - 4)$ | $0 \le x \le 1$ |
| 3 | $f_{LF_1}(x) = 0.5 f_{HF}(x) + 10(x - 0.5) + 5$ <br> $f_{LF_2}(x) = 0.4 f_{HF}(x) - x - 1$ | $f_{HF}(x) = (6x - 2)^2 \sin(12x - 4)$ | $0 \le x \le 1$ |
| 4 | $f_{LF_1}(x) = \frac{(x-0.5)(x-4)(x-9)}{20} + 2$ <br> $f_{LF_2}(x) = \sin(x) + 0.2x + 0.5$ | $f_{HF}(x) = \sin(x) + 0.2x + \frac{(x-5)^2}{16} + 0.5$ | $0 \le x \le 10$ |
| 5 | $f_{LF_1}(x) = x(x - 5)(x - 12)/30$ <br> $f_{LF_2}(x) = (x + 2)(x - 5)(x - 10)/30$ | $f_{HF}(x) = \sin\left(\frac{\pi x}{5}\right)$ | $0 \le x \le 10$ |
| 6 | $f_{LF_1}(\boldsymbol{x}) = f_{HF}(0.7x_1, 0.7x_2) + x_1 x_2 - 65$ <br> $f_{LF_2}(\boldsymbol{x}) = f_{HF}(0.8x_1, 0.6x_2) + x_1^4 + 32$ | $f_{HF}(\boldsymbol{x}) = \quad 4x_1^2 - 2.1x_1^4 + \frac{x_1^6}{3} + \dots$ <br> $+ x_1 x_2 - 4 + 4x_2^4$ | $\boldsymbol{x} \in [-2, 2]^2$ |
| 7 | $f_{LF_1}(\boldsymbol{x}) = \quad -\sin x_1 - e^{\frac{x_1}{100}} + 10.3 + \dots$ <br> $+ 0.03(x_1 - 0.3)^2 + (x_2 - 1)^2/10$ <br> $f_{LF_2}(\boldsymbol{x}) = -\sin 0.9x_1 - e^{\frac{0.9x_1}{100}} + 10 + 0.64x_2^2/10$ | $f_{HF}(\boldsymbol{x}) = -\sin x_1 - e^{\frac{x_1}{100}} + 10 + x_2^2/10$ | $\boldsymbol{x} \in [0, 1]^2$ |
| 8 | $f_{LF_1}(\boldsymbol{x}) = 0.79\left(1 + \frac{\sin x_1}{10}\right) f_{HF}(\boldsymbol{x}) - 2x_1 + x_2^2 + x_3^2 + 0.5$ <br> $f_{LF_2}(\boldsymbol{x}) = f_{HF}(\boldsymbol{x}) + e^{x_3/2} - x_1/10$ | $f_{HF}(\boldsymbol{x}) = \frac{x_1}{2}\left(\sqrt{1 + (x_1 + x_3^2)x_4/x_1^{20}}\right)$ | $\boldsymbol{x} \in [0.5, 1]^4$ |

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
