# Peer review of "Extended Hierarchical Kriging Method for Aerodynamic Model Generation Incorporating Multiple Low-Fidelity Datasets"

_aerospace, doi:10.3390/aerospace11010006_

Round 1

Reviewer 1 Report

Comments and Suggestions for Authors

All in all, it is a good job. Congratulations!

Detailed comments: in order not to confuse the symbol for the exponential exp, the logarithm ln and sin, it is proposed that they should always be written in roman type and not in italics. Revise expressions 15, 16, 19, 20, 24, 25, 26 and table 2 and B1.

Reviewer 2 Report

Comments and Suggestions for Authors

In this manuscript, the authors proposed the Extended Hierarchical Kriging (EHK) method, which leverages a unique Bayesian-based MFSM framework, simultaneously combining non-level LF models using scaling factors to construct a global trend model. This research is interesting and has practical significance. Some of the suggestions and questions are listed in the attached file.
